# Function-to-Style Guidance of LLMs for Code Translation

**Longhui Zhang** [1]  **Bin Wang** [1]  **Jiahao Wang** [1]  **Xiaofeng Zhao** [2]  **Min Zhang** [2]  **Hao Yang** [2]
**Meishan Zhang** [1]  **Yu Li** [3]  **Jing Li**✉ [1]  **Jun Yu** [1]  **Min Zhang** [1]

## Abstract

Large language models (LLMs) have made significant strides in code translation tasks. However, ensuring both the correctness and readability of translated code remains a challenge, limiting their effective adoption in real-world software development. In this work, we propose F2STRANS, a *function-to-style guiding paradigm* designed to progressively improve the performance of LLMs in code translation. Our approach comprises two key stages: (1) *Functional learning*, which optimizes translation correctness using high-quality source-target code pairs mined from online programming platforms, and (2) *Style learning*, which improves translation readability by incorporating both positive and negative style examples. Additionally, we introduce a novel code translation benchmark that includes up-to-date source code, extensive test cases, and manually annotated ground-truth translations, enabling comprehensive functional and stylistic evaluations. Experiments on both our new benchmark and existing datasets demonstrate that our approach significantly improves code translation performance. Notably, our approach enables Qwen$_{1.5B}$ to outperform prompt-enhanced Qwen$_{32B}$ and GPT-4 on average across 20 diverse code translation scenarios.

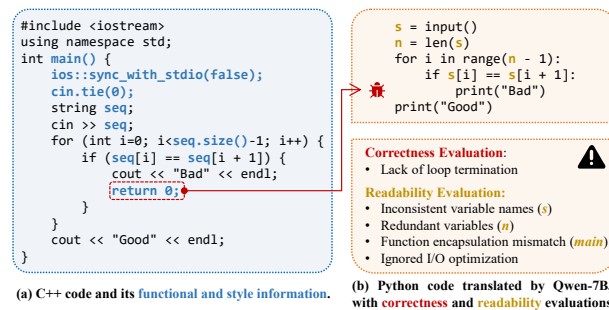

(a) C++ code and its functional and style information.

(b) Python code translated by Qwen-7B, with correctness and readability evaluations.

Figure 1: Limitations in the correctness and readability of code translation by LLMs.

## 1. Introduction

Code translation involves converting code from one programming language to another, a task usually required for application porting or software migration (Nguyen et al., 2013). Traditionally, this task primarily relied on rule-based methods that required skilled programmers to handle com-

plex cases manually (Zhong et al., 2010). The advent of deep learning has led to the development of various learning-based strategies. A prominent approach involves pretraining on monolingual code datasets, followed by fine-tuning with bilingual corpora to improve translation accuracy (Lachaux et al., 2020; Wang et al., 2021). Although learning-based techniques have significantly outperformed traditional rule-based methods, their performance remains insufficient for real-world deployment (Yang et al., 2024).

Large language models (LLMs), such as GPT-4 (Achiam et al., 2023), have revolutionized various coding tasks, including code generation (Liu et al., 2024a) and program repair (Fan et al., 2023), with their remarkable performance. This success has spurred growing interest among researchers in leveraging LLMs for code translation (Yan et al., 2023b). Simple prompt-based learning allows LLMs to translate code effectively, and optimized prompts can further enhance their performance. For example, Bhattarai et al. (2024a;b) improved translation accuracy using a retrieval-augmented generation (RAG) strategy. Yang et al. (2024); Pan et al. (2024) utilized compiler feedback to iteratively refine translations. Powered by LLMs, current code translation models have reached unprecedented levels of performance, surpassing traditional approaches (Tao et al., 2024).

While the effectiveness of LLMs in code translation tasks is widely acknowledged, most models face two critical limitations, as illustrated in Figure 1: (i) **Correctness**: For instance, StarCoder$_{3B}$ achieves an average success rate of only 7% on the traditional code translation benchmark Co-

[1]Harbin Institute of Technology, Shenzhen, China. [2]Huawei Translation Services Center, Beijing, China. [3]Zhejiang University, Hangzhou, China. Correspondence to: Jing Li <jingli.phd@hotmail.com>.

*Proceedings of the 42$^{nd}$ International Conference on Machine Learning*, Vancouver, Canada. PMLR 267, 2025. Copyright 2025 by the author(s).

deNet (Pan et al., 2024). (ii) **Readability**: Even when translations are functionally correct, they often fail to preserve the source code's style, including code structure and variable naming conventions (Weisz et al., 2022). This lack of readability imposes a substantial burden on developers, as reading poorly structured code often takes longer than writing it from scratch (Martin, 2009). Undeniably, more powerful LLMs can produce higher-quality code translations, but they come with notable drawbacks, such as massive model sizes (*e.g.,* Qwen$_{32B}$ (Qwen et al., 2025)) or limited accessibility (*e.g.,* GPT-4). Relying solely on the inherent capabilities of LLMs to overcome these issues is only a short-term solution.

To address these challenges, we propose F2STRANS, a two-stage guidance framework to improve both correctness and readability: *functional learning* followed by *style learning*. Functional learning involves training the model to generate target code that preserves the functionality of the source code. To achieve this, F2STRANS mines cross-language code pairs from online programming platforms, selects pairs with consistent solutions and functionality, and utilizes these pairs for functional learning. Style learning ensures the model accurately preserves the stylistic features of the source code in the translated target code, thereby improving readability. In this stage, F2STRANS generates target code samples exhibiting both good and poor stylistic quality, enabling the model to recognize and prioritize maintaining stylistic consistency through the style learning.

We conduct extensive experiments to validate our approach. To overcome the limitations of existing benchmarks, such as outdated source code, insufficient test cases, and missing ground-truth translations, we construct a new benchmark. Utilizing this new benchmark, along with the traditional benchmarks, we evaluate F2STRANS across 20 code translation scenarios encompassing five programming languages: C, C++, Go, Java, and Python. The results demonstrate that our approach is effective across LLMs of varying types and scales, including StarCoder$_{3B}$ and Qwen$_{0.5-7B}$. Notably, with our approach, Qwen$_{1.5B}$ surpasses GPT-4 in code translation tasks.

We summarize the key contributions of our work as follows:

- We propose F2STRANS, a function-to-style guidance framework designed to enhance both the correctness and readability of code translations generated by LLMs.

- We create a comprehensive benchmark to rigorously evaluate the functional accuracy and stylistic consistency of code translations.

- Our approach significantly improves the quality of translated code across various types and sizes of LLMs in 20 investigated translation scenarios.

## 2. Methodology

As illustrated in Figure 2, F2STRANS employs a two-stage progressive learning paradigm. First, to ensure functionally consistent code translation, we perform a functional learning using function-consistent code pairs. Second, to improve the style alignment between source and target code, we propose a novel style-learning mechanism based on positive and negative style translation examples.

### 2.1. Function-oriented Guidance

A fundamental requirement of code translation is functional consistency—ensuring that source and target code produce identical outputs for the same inputs. A simple strategy (Khan et al., 2024) is to fine-tune LLMs using source and target language code that solves the same programming problem on online programming platforms. Platforms such as Codeforces host numerous programming problems, along with test cases and code solutions submitted by developers worldwide, making this strategy feasible.

However, this strategy can create ambiguity in the model's understanding of the optimal translation, since programming problems typically have multiple solutions. More critically, solution inconsistencies may lead to divergent output behaviors between technically correct code solutions. For example, while both Dijkstra's and Floyd-Warshall's algorithms solve the shortest path problem, they may produce different but equally valid results for the same graph. Obviously, training on code pairs with inconsistent outputs is sub-optimal. These limitations motivate us to enhance the quality of the data generated by this strategy.

**Function-consistent Data Construction.** To develop our dataset, we first select highly relevant source-target code pairs from a large number of solutions to the same programming problem, and then retain those exhibiting identical input-output behavior.

▶ *Relevance-driven Code Pair Selection.* To balance effectiveness and efficiency, we employ a two-step method to identify highly relevant code pairs. First, we use a lightweight code embedding model Jina (Günther et al., 2024) to retrieve the top ten similar code pairs based on cosine similarity of embeddings. Next, we apply a rating scale–based method where an LLM judge assesses the solution consistency of code pairs using fine-grained labels. The scoring process is defined as:

$$score\left(src, tgt\right) = \sum_{k} p\left(k \mid \mathcal{P}_s(src, tgt)\right) \cdot k$$
$$p\left(k \mid \mathcal{P}_s(src, tgt)\right) = \frac{\exp\left(s_k\right)}{\sum_{k'} \exp\left(s_{k'}\right)}, \quad (1)$$

where we use Qwen$_{7B}$ as the LLM judge. Here, $src$ and $tgt$ represent source and target code, respectively, while

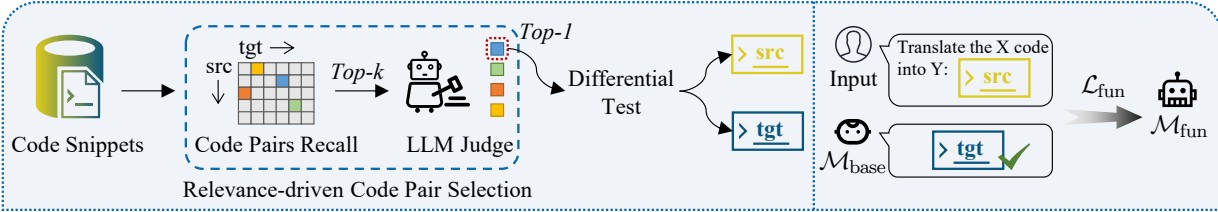

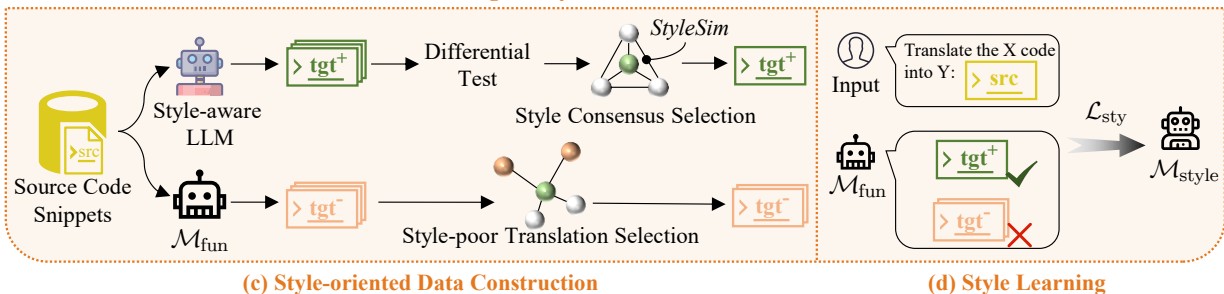

Figure 2: Overview of our F2STRANS. Firstly, the base LLM $\mathcal{M}_{\text{base}}$ is transformed into an intermediate model $\mathcal{M}_{\text{fun}}$ after function-oriented guidance. Subsequently, $\mathcal{M}_{\text{fun}}$ is refined into the final model $\mathcal{M}_{\text{sty}}$ through style-oriented guidance.

$\mathcal{P}_s(src, tgt)$ is the prompt for the LLM judge, shown in Appendix F.1. The variable $k$ is an integer ranges from 1 to $K$, with higher values signifying greater relevance, and $s_k$ is the log-likelihood score of label $k$ generated by the LLM judge when prompted with $\mathcal{P}_s(src, tgt)$. We use fine-grained labels $\{1, \ldots, K\}$ instead of binary labels, because code pairs for the same programming problem often exhibit varying degrees of solution similarity. Besides, to ensure each code pair receives a distinct score, we aggregate the log-likelihood scores $s_*$ of all labels to obtain continuous relevance scores.

▶ *Differential Testing.* To ensure functional consistency, we perform differential testing (McKeeman, 1998) on the most relevant code pairs. This process involves executing identical input on both the source and target code and comparing their outputs for discrepancies. Only code pairs whose input and output behavior are exactly the same are retained.

**Functional Learning.** Instruction Fine-tuning (IFT) involves training LLMs on instruction-output pairs with a next-token prediction objective to enhance the models' ability to follow instructions (Zhang et al., 2024). We preliminary improve the code translation performance of base LLM $\mathcal{M}_{\text{base}}$ by applying IFT on the constructed function-consistent code translation data as follow:

$$\mathcal{L}_{\text{fun}}(scr, tgt) = -\sum_i \log p\left(tgt_i \mid \mathcal{P}(src), tgt_{<i}\right), \quad (2)$$

where $tgt_i$ denotes the $i$-th token of target code $tgt$, $tgt_{<i}$ represents the token sequence preceding the $i$-th token in $tgt$, and $\mathcal{P}(src)$ is a code translation prompt designed to

translate the source code $src$. The trained model is $\mathcal{M}_{\text{fun}}$.

## 2.2. Style-oriented Guidance

Although our functional guidance ensures data quality at the solution and function levels, style inconsistencies inevitably persist in code pairs derived from online programming platforms. These inconsistencies usually include variations in variable naming, function signatures, code structure, and comments between source and target code. Training on such style-inconsistent data can limit the model's learning of the code style, thereby diminishing the readability of the translated code. Therefore, we introduce style-oriented guidance to alleviate this issue.

**Style-oriented Data Construction.** We construct positive and negative code translation data to help $\mathcal{M}_{\text{fun}}$ discern the desired code style through comparison. Positive translations maintain stylistic consistency with the source code, while negative translations do not.

▶ *Positive Translation Construction.* We use a strong LLM Qwen$_{32B}$ to generate multiple style-aware translations and then select the optimal translation from these candidates. Specifically, we first design a style-aware prompt, as detailed in Appendix F.1, which explicitly instructs the LLM to adhere to the stylistic conventions of the source code. Using this prompt, Qwen$_{32B}$ generates $m$ translations of the source code. Next, we perform differential testing to filter these translations, retaining only the subset $T^+$ that successfully passes all test cases, thereby ensuring functional correctness. Finally, we select the target code $tgt^+$

that best preserves the source's style from $T^+$.

A major challenge in this process is how to choose the best translation $tgt^+$ from $T^+$. While our style-aware prompt contributes to good code style, the occasional style error is still inevitable. Existing style evaluation methods, such as CSSim (Li et al., 2024), are limited to code pairs within the same language. To avoid the use of abnormal code translation, we propose a style consensus selection mechanism, which identifies the optimal translation by selecting the one with the highest stylistic similarity to the other candidate translations. The selection mechanism can be mathematically represented as follow:

$$tgt^+ = \arg\max_{tgt_i \in T^+} \sum_{tgt_j \in T^+, i \neq j} \text{StyleSim}\left(tgt_i, tgt_j\right), \quad (3)$$

where $\text{StyleSim}(.)$ is a function that measures the stylistic similarity between two code snippets in the target language. We use the CSSim metric (Li et al., 2024) as StyleSim, which quantifies code stylistic similarity based on variable naming, API invocation, and code structure. A detailed explanation of CSSim can be found in Appendix A.

▶ *Negative Translation Collection Construction.* To ensure that negative translations are representative, we use $\mathcal{M}_{fun}$ as the negative translator. Specifically, $\mathcal{M}_{fun}$ first generate multiple candidate negative translations. Then, we evaluate their style consistency with the positive translations $tgt^+$ using the CSSim metric and retain $n$ negative translations $T^-$ whose CSSim values are less than $\alpha$.

**Style Learning.** Inspired by contrastive learning (Chen et al., 2020), we propose a list-wise loss function to encourage $\mathcal{M}_{\text{fun}}$ generate style-consistent target code $tgt^+$ while suppressing inconsistent translations $T^-$:

$$\mathcal{L}_{\text{list}}(src, tgt^+, T^-) = -\log \frac{\exp(\mathcal{S}(src, tgt^+))}{\sum_{tgt \in T} \exp(\mathcal{S}(src, tgt))}$$
$$\mathcal{S}(src, tgt) = \prod_j p\left(tgt_j \mid \mathcal{P}(src), tgt_{<j}\right), \quad (4)$$

where $\mathcal{S}(src, tgt)$ denotes the probability that $\mathcal{M}_{\text{fun}}$ translates the source code $src$ into the target code $tgt$, and $T = T^- \cup \{tgt^+\}$. Additionally, we apply IFT on positive translations to emphasize their importance:

$$\mathcal{L}_{\text{sty}} = \beta \cdot \mathcal{L}_{\text{list}}(scr, tgt^+, T^-) + (1 - \beta) \cdot \mathcal{L}_{\text{ift}}(scr, tgt^+), \quad (5)$$

where $\beta$ is a trade-off hyperparameter ranging between 0 and 1, and $\mathcal{L}_{\text{ift}}$ is the IFT loss, calculated the same way as $\mathcal{L}_{\text{fun}}$ in Eq. 2.

## 3. Experiments

### 3.1. Benchmark Construction

Since pre-training LLMs requires handling vast datasets, traditional benchmarks face the risk of data leakage even

|  | #Lang | #Code | Date | #Cases | GT |
|---|---|---|---|---|---|
| CodeNet | 5 | $200 \times 20$ | Pre-2021 | $1 \times 20$ | ✗ |
| F2STRANS (Ours) | 5 | $1000 \times 20$ | Mid-2024 | $50 \times 20$ | ✓ |

Table 1: Comparison of CodeNet and our evaluation benchmark. Both datasets cover 20 translation scenarios across five languages (Lang). Our benchmark surpasses CodeNet in terms of a larger and more up-to-date codebase (Code and Date), extensive test cases (Cases), and manually annotated ground-truth translations (GT).

after meticulous data cleaning, leading to inaccurate evaluations (Xu et al., 2024). For example, the latest data for the CodeNet benchmark comes from 2020 (Puri et al., 2021). To accurately assess the code translation capabilities of LLMs, a benchmark based on more recent data is essential.

Motivated by this insight, we introduce a new code translation benchmark with three key advantages over traditional benchmarks: (i) **Up-to-date source code**: We first collect the most recently released programming problems from Codeforces, and then select up to two code solutions from each problem as source code. (ii) **Extensive test cases**: Each source code undergoes extensive manually-annotated test cases that encompass edge conditions and difficult scenarios of programming problems. (iii) **Consistent functional and stylistic translations**: We manually translate each source code into the target language to support functional and stylistic evaluation of the translated code. The detailed statistics presented in Table 1.

### 3.2. Experimental Settings

**Implementation Details.** In function-oriented guidance, we set the maximum algorithmic consistency label $K$ in Eq. 1 to 5. In the style-oriented guidance, we set both the numbers of positive translations $T^+$ and negative translations $T^-$, namely $m$ and $n$, to 10, with the value of $\alpha$ in negative translation collection construction set to 0.8 and the trade-off hyperparameter $\beta$ in Eq. 5 fixed at 0.6. We combine training data from various translation scenarios for mixed training, enabling a single LLM to translate among all the investigated languages. The LLM-judge prompt $\mathcal{P}_s$ in function-oriented guidance, the style-aware prompt in style-oriented guidance, and the prompt $\mathcal{P}$ of F2STRANS in Eq. 2 and 4 are shown in Appendix F.1. Appendix B shows more detailed implementation details.

**Baselines and Metric.** We evaluate our approach against two SOTA LLMs: $\text{Qwen}_{32B}$ and GPT-4. For each model, we implement four established prompt learning strategies: Direct Prompt Learning (Yang et al., 2024), Chain of Thought (CoT) (Yan et al., 2023b), RAG (Bhattarai et al., 2024a;b), Self-Debug Prompt Learning (Yang et al., 2024; Pan et al., 2024). These strategies have been proposed and

| Method | LLM | Translation C → {} | | | | Translation C++ → {} | | | | Translation Go → {} | | | | Translation Java → {} | | | | Translation Py → {} | | | | Avg. |
|---|---|---|---|---|---|---|---|---|---|---|---|---|---|---|---|---|---|---|---|---|---|---|
| | | C++ | Go | Java | Py | C | Go | Java | Py | C | C++ | Java | Py | C | C++ | Go | Py | C | C++ | Go | Java | |
| | | | | | | | | | (I) CodeNet Benchmark | | | | | | | | | | | | | |
| Direct | Qwen32B | 88.0 | 68.9 | 76.4 | 61.3 | 83.5 | 69.0 | 81.0 | 59.5 | 69.4 | 74.9 | 75.4 | 58.3 | 71.2 | 79.3 | 64.2 | 61.1 | 76.9 | 81.4 | 69.9 | 88.5 | 72.9 |
| CoT | | 82.9 | 72.4 | 75.4 | 62.8 | 80.5 | 61.5 | 81.0 | 65.5 | 75.9 | 72.4 | 77.9 | 63.3 | 78.3 | 80.3 | 58.6 | 61.6 | 73.4 | 82.4 | 76.4 | 88.4 | 73.5 |
| RAG | | 87.5 | 76.5 | 79.0 | 70.1 | 83.1 | 73.1 | 81.7 | 65.6 | 69.0 | 75.4 | 76.1 | 66.0 | 77.4 | 78.4 | 71.9 | 66.3 | 84.6 | 81.5 | 77.0 | 87.9 | 76.4 |
| Self-debug | | 89.1 | 74.4 | 77.9 | 70.1 | 86.6 | 71.7 | 83.9 | 67.3 | 74.3 | 78.4 | 79.9 | 68.0 | 76.6 | 82.5 | 68.8 | 69.4 | 81.0 | 85.0 | 76.0 | 90.5 | 77.6 |
| Direct | GPT-4 | 91.0 | 69.5 | 80.9 | 62.6 | 83.3 | 68.2 | 84.0 | 64.0 | 77.4 | 80.4 | 77.9 | 69.4 | 77.8 | 83.9 | 66.7 | 61.3 | 76.9 | 84.2 | 58.1 | 82.3 | 75.3 |
| CoT | | 88.8 | 73.2 | 81.1 | 65.2 | 82.7 | 67.0 | 83.7 | 68.8 | 81.3 | 80.8 | 80.6 | 73.5 | 86.3 | 86.2 | 63.5 | 67.1 | 73.7 | 84.5 | 62.6 | 83.5 | 76.7 |
| RAG | | 91.2 | 76.2 | 85.3 | 72.0 | 84.6 | 70.6 | 84.7 | 76.2 | 78.3 | 77.0 | 72.6 | 77.7 | 83.2 | 84.1 | 72.9 | 72.1 | 84.4 | 86.3 | 64.3 | 85.1 | 78.9 |
| Self-debug | | 93.5 | 72.5 | 87.5 | 78.5 | 86.5 | 72.0 | 87.5 | 74.0 | **84.5** | **88.0** | **88.5** | 81.0 | 87.0 | 89.5 | 70.0 | 78.5 | 85.5 | 87.0 | 62.0 | 85.0 | 81.9 |
| **F2STRANS (Ours)** | Qwen0.5B | 91.0 | 78.9 | 81.9 | 73.9 | 84.5 | 82.0 | 84.5 | 75.5 | 70.9 | 73.9 | 77.4 | 76.4 | 74.2 | 85.4 | 77.3 | 75.8 | 74.4 | 80.9 | 73.4 | 85.4 | 78.9 |
| | Qwen1.5B | 93.0 | 83.4 | 89.4 | 83.9 | 90.5 | **90.0** | 87.5 | 80.0 | 79.4 | 78.4 | 82.4 | 83.9 | 85.9 | 86.9 | 86.4 | 82.3 | 84.4 | 87.4 | 78.9 | 87.4 | 85.1 |
| | Qwen3B | **93.6** | 86.4 | 90.5 | 82.9 | 90.0 | 89.0 | **93.5** | 84.0 | 79.4 | 81.4 | 86.4 | 83.9 | 85.9 | 88.9 | 85.9 | 84.8 | 89.9 | 88.4 | 85.4 | 91.0 | 87.1 |
| | Qwen7B | 91.5 | **92.5** | **93.5** | 86.5 | **92.0** | **92.5** | 93.0 | **88.0** | 82.9 | 86.9 | 89.4 | 87.9 | **88.4** | **89.9** | **88.4** | **87.4** | **93.0** | **89.4** | 87.4 | **94.5** | **89.8** |
| | StarCoder3B | 92.0 | 87.9 | 89.9 | 84.4 | **92.0** | 89.5 | 91.5 | 83.5 | 80.4 | 81.9 | 84.9 | 84.9 | 86.4 | 85.4 | 83.8 | 85.4 | 86.9 | 85.9 | 84.9 | 92.5 | 86.7 |
| | | | | | | | | | (II) F2STRANS Benchmark (Ours) | | | | | | | | | | | | | |
| Direct | Qwen32B | 81.7 | 44.4 | 68.5 | 48.5 | 57.4 | 33.6 | 56.4 | 39.1 | 63.2 | 69.9 | 61.3 | 48.7 | 71.6 | 77.1 | 48.1 | 63.1 | 32.7 | 37.1 | 32.1 | 43.5 | 53.9 |
| CoT | | 80.3 | 43.7 | 68.2 | 51.9 | 54.4 | 34.3 | 56.0 | 42.0 | 66.4 | 72.7 | 64.9 | 53.8 | 68.0 | 78.5 | 44.4 | 63.4 | 29.9 | 40.9 | 37.0 | 45.5 | 54.8 |
| RAG | | 81.3 | 57.9 | 67.6 | 57.2 | 56.6 | 46.6 | 55.3 | 41.6 | 61.4 | 69.3 | 60.9 | 54.6 | 71.1 | 77.3 | 59.9 | 69.2 | 34.3 | 37.8 | 34.1 | 44.0 | 56.9 |
| Self-debug | | 83.1 | 49.0 | 69.8 | 56.0 | 60.4 | 35.7 | 59.0 | 47.2 | 68.5 | 72.2 | 65.8 | 57.9 | 77.8 | 80.1 | 51.3 | 71.8 | 35.3 | 39.4 | 37.3 | 46.0 | 58.2 |
| Direct | GPT-4 | 89.9 | 58.1 | 81.3 | 51.4 | 68.9 | 52.6 | 70.7 | 52.6 | 70.9 | 81.0 | 74.1 | 68.9 | 77.2 | 87.7 | 60.9 | 62.4 | 34.4 | 44.4 | 38.3 | 46.7 | 63.6 |
| CoT | | 88.2 | 60.2 | 80.8 | 54.0 | 69.2 | 50.2 | 71.2 | 56.5 | 75.1 | 82.2 | 75.7 | 72.7 | 81.5 | 89.2 | 57.4 | 63.1 | 34.2 | 42.8 | 40.5 | 44.5 | 64.5 |
| RAG | | 89.7 | 61.4 | 87.0 | 59.7 | 69.2 | 56.0 | 72.2 | 61.8 | 71.4 | 80.8 | 73.4 | 75.1 | 81.0 | 87.0 | 64.6 | 67.5 | 39.5 | 47.6 | 42.5 | 49.3 | 66.8 |
| Self-debug | | 91.3 | 61.5 | 85.6 | 63.2 | 71.4 | 55.7 | 74.1 | 61.4 | 72.4 | 83.6 | 78.7 | 76.7 | **84.5** | 88.7 | 67.4 | 72.9 | 40.6 | 46.8 | 42.8 | **51.7** | 68.5 |
| **F2STRANS (Ours)** | Qwen0.5B | 91.0 | 71.5 | 76.1 | 64.8 | 66.0 | 59.1 | 60.6 | 53.2 | 64.8 | 74.8 | 67.3 | 69.2 | 70.3 | 84.0 | 73.9 | 72.8 | 25.1 | 35.1 | 27.6 | 34.1 | 62.1 |
| | Qwen1.5B | 93.5 | 75.7 | 83.6 | 71.3 | 75.1 | 68.2 | 71.2 | 62.3 | 70.7 | 79.9 | 73.2 | 76.6 | 79.5 | 89.2 | 83.7 | 82.1 | 35.1 | 44.8 | 35.7 | 44.6 | 69.8 |
| | Qwen3B | **95.2** | 81.3 | 86.7 | 76.1 | 78.2 | 71.8 | 75.4 | 68.9 | 75.2 | 82.5 | 77.4 | 80.7 | 81.1 | **91.1** | 86.8 | 84.6 | 38.9 | 48.4 | 43.6 | 50.4 | 73.7 |
| | Qwen7B | 94.6 | **84.0** | 87.6 | 76.1 | **79.5** | 75.8 | **77.6** | **69.4** | 76.4 | **85.1** | **85.1** | **82.0** | 82.7 | 91.0 | **88.2** | **84.7** | **43.2** | **56.5** | **47.5** | 51.3 | **75.9** |
| | StarCoder3B | 94.3 | 78.8 | 86.5 | **76.3** | 77.8 | 71.0 | 75.1 | 66.7 | **76.6** | 82.0 | 77.2 | 80.2 | 82.0 | 90.2 | 87.0 | 83.4 | 37.9 | 45.2 | 42.9 | 46.5 | 72.9 |

Table 2: Code translation results of various models on CodeNet and our benchmark. Following standard practice, we adopt CA as the evaluation metric. The bold values represent the best results, while the underlined values indicate the second-best.

validated in related works to enhance model performance in code translation tasks. Detailed descriptions of each strategy are provided in Appendix C.

Following standard practice (Lachaux et al., 2020), we adopt Computational Accuracy (CA) as our evaluation metric, which measures the proportion of translated code that produce identical execution results to the source code across all inputs. We allowed each LLM only one translation attempt per source code. However, for self-debug prompt learning, which inherently requires multiple LLM invocations, we allow an additional iteration for bug fixing.

### 3.3. Main Results

To verify the broad applicability of F2STRANS, our experiments involve various types and sizes of LLMs, including Qwen0.5−7B (Qwen et al., 2025) and StarCoder3B (Lozhkov et al., 2024). The performance of our model and baselines on CodeNet and our benchmark is presented in Table 2, and additional benchmark results in Appendix D.

**Evaluation on CodeNet Benchmark.** From Table 2(I),

we recognize the immense potential of LLMs in code translation tasks. For instance, F2STRANS enables Qwen0.5B to surpass RAG-based Qwen32B in average performance across the various translation scenarios examined. While F2STRANS lags behind GPT-4 with self-debug prompt learning in some specific cases, such as Go-to-C translation, it is important to consider the increased computational cost associated with the additional reasoning steps required by self-debug prompt learning for bug fixing. Therefore, the advantages of F2STRANS remain significant.

**Evaluation on F2STRANS Benchmark.** As shown in Table 2(II), all evaluated models score at least 10 points lower on average in our benchmark compared to CodeNet, highlighting the greater complexity of our proposed benchmark. An interesting observation is the significantly weaker performance of all models on Python translation within our benchmark. This may be attributed to Python's interpreted nature, contrasting with the compiled nature of the other languages. This disparity in language types introduces potential challenges in code translation. Nonetheless, the overall trend observed on our benchmark is consistent with that of the CodeNet benchmark. F2STRANS enables

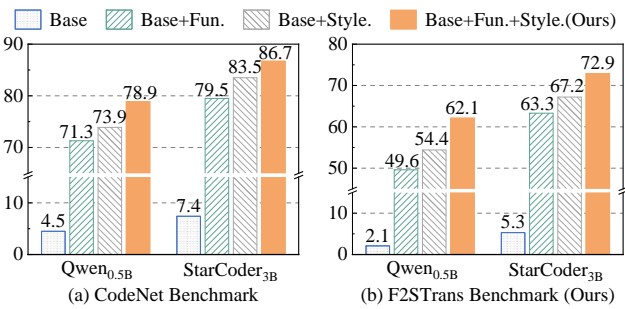

Figure 3: Model performance under different training strategies: base LLMs, function guidance (Fun.), style guidance (Style.), and our function-to-style guidance.

| Method | $\text{Qwen}_{0.5B}$ | | $\text{StarCoder}_{3B}$ | |
|---|---|---|---|---|
| | CodeNet | Ours | CodeNet | Ours |
| F2STRANS (Ours) | **78.9** | **62.1** | **86.7** | **72.9** |
| ● *Function-oriented Guidance* | | | | |
| w/o RdSel | 75.5 | 57.5 | 83.7 | 68.8 |
| w/o LLM Judge | 77.5 | 59.5 | 84.3 | 70.2 |
| w/o Dif. Test | 78.2 | 60.2 | 85.8 | 71.0 |
| ● *Style-oriented Guidance* | | | | |
| w/o StyPro | 78.6 | 61.5 | 86.3 | 72.4 |
| w/o SCS | 77.7 | 58.6 | 84.2 | 69.4 |
| w/o SpTS | 78.4 | 60.3 | 86.3 | 71.4 |
| w/o $\mathcal{L}_{\text{list}}$ | 74.3 | 56.8 | 84.1 | 68.8 |
| w/o $\mathcal{L}_{\text{ift}}$ | 75.9 | 59.0 | 85.4 | 70.8 |

Table 3: Ablation results of our F2STRANS, evaluating the contribution of the following components: relevance-driven code pair selection (RdSel), LLM judge and differential test (Dif. Test) in function guidance, along with style-aware prompt learning (StyPro.), style consensus selection in Eq. 3 (SCS), style-poor translation selection (SpTS) and the loss function in style guidance.

## 3.4. Ablation Analysis

Our ablation analysis quantifies the contribution of each part of F2STRANS to the overall performance improvements.

**Function-to-Style Guidance.** Figure 3 shows the individual contributions of function and style guidance based on $\text{Qwen}_{0.5B}$ and $\text{StarCoder}_{3B}$. As shown, both training stages exhibit significant performance gains, validating their importance to optimal final results. In particular, the influence of style guidance is highly remarkable. This is unsurprising, as style guidance involves both positive and negative translations, and the positive translations are derived from $\text{Qwen}_{32B}$ rather than online data. Without these two stages, both $\text{Qwen}_{0.5B}$ and $\text{StarCoder}_{3B}$ score below 10, underscoring the inherent limitations of naive LLMs in code translation tasks.

**Function-oriented Guidance.** The first part of Table 3 examines the impact of three modules in this stage:

▶ *Relevance-driven Code Pair Selection.* To validate the effectiveness of this module, we compared F2STRANS to a variant that randomly selects code pairs, disregarding solution differences. The notable performance decline observed with this variant confirms our hypothesis that the solution gaps between code versions are crucial and cannot be ignored. Bridging these gaps is essential for achieving optimal model performance.

▶ *LLM Judge.* To assess the impact of LLM judgments, we implemented a variant that directly uses the embedding model Jina to select the top-ranked relevant code pair for training. The observed performance decline in this model variant indicates that, although Jina is specifically designed for relevance assessment, LLM judgments can effectively refine its retrieval results. This further corroborates the

strong task generalization capabilities of LLMs.

▶ *Differential Testing.* Although we have strived to maximize the solution relevance of the training data, overlooking differential testing has undeniably adversely affected the training process. This phenomenon indicates that assessing the input–output behaviors of code pairs is the most effective approach to verifying their functional consistency.

**Style-oriented Guidance.** As shown in Table 3 and Figure 4, we evaluate four key modules of style supervision:

▶ *Style-aware Prompt Learning.* We employ style-aware prompts to guide LLMs in generating positive translations that align with the source code style. To evaluate the necessity of this configuration, we remove all style-related information from the prompt, thereby allowing the LLMs to generate code without stylistic constraints. As shown in Table 3 under "w/o StyPro", both $\text{Qwen}_{0.5B}$ and $\text{StarCoder}_{3B}$ exhibit decreased performance, indicating that well-defined styles in the training data facilitate LLMs' adaptation to code translation tasks.

▶ *Style-oriented Data Selection of Positive and Negative Candidate Translations.* As illustrated in Figure 2, we employ a style consensus strategy to select the optimal positive translation from those generated by $\text{Qwen}_{32B}$ and utilize negative translations that significantly differ in style from the positive data. In the ablation studies presented under the "w/o SCS" and "w/o SpTS" entries in Table 3, we omit these two data selection methods and instead randomly select one positive candidate translation and $n$ negative candidate translations for the style learning in Eq. 5.

The results demonstrate that random data is of inferior qual-

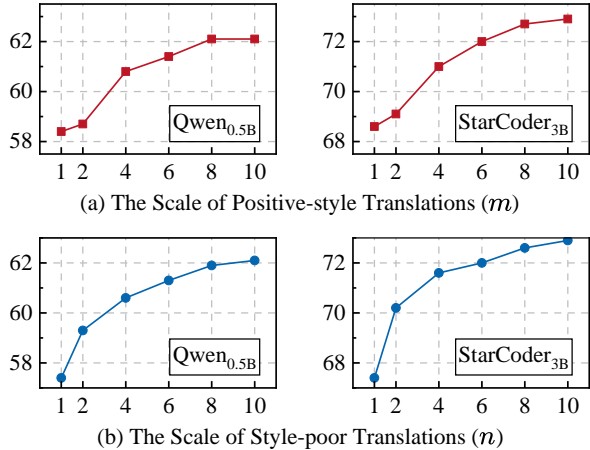

(a) The Scale of Positive-style Translations ($m$)

(b) The Scale of Style-poor Translations ($n$)

Figure 4: Impact of the number of positive translations generated by Qwen$_{32B}$ and the number of negative translations used in style learning, based on our benchmark.

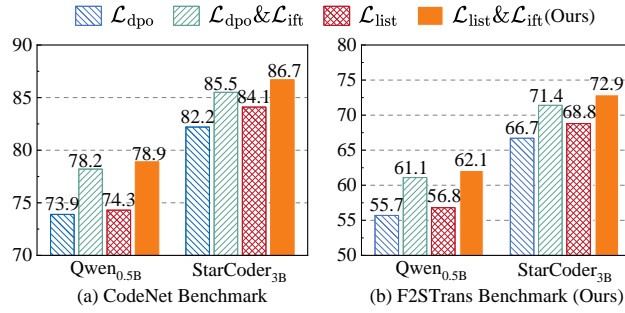

(a) CodeNet Benchmark

(b) F2STrans Benchmark (Ours)

Figure 5: Results of style learning based on various loss functions.

ity compared to our carefully curated dataset. This yields two key insights: (I) Even functionally correct translations generated by powerful LLMs may contain subtle stylistic inconsistencies. (II) Ensuring a clear stylistic distinction between positive and negative translations facilitates more effective style learning.

▶ *The Number of Positive and Negative Candidate Translations.* Figure 4 illustrates the influence of these two numbers, $m$ and $n$, based on our benchmark. We observe that increasing both $m$ and $n$ significantly enhances model performance, with results stabilizing around $m = 8$ and $n = 10$. This is expected, as a larger number of positive candidates increases the likelihood of identifying the optimal translation, while more negative translations expose potential deficiencies of $\mathcal{M}_{\text{fun}}$. Furthermore, the steeper curves demonstrate that scaling these data yields greater benefits for StarCoder$_{3B}$, indicating that larger model sizes achieve higher performance ceilings.

▶ *Loss Function.* Our loss function comprises $\mathcal{L}_{\text{list}}$ and $\mathcal{L}_{\text{ift}}$. Ablation studies in Table 3 indicate that $\mathcal{L}_{\text{list}}$ significantly enhances model performance, and the addition of $\mathcal{L}_{\text{ift}}$ further increases these benefits. Moreover, $\mathcal{L}_{\text{list}}$ outperforms $\mathcal{L}_{\text{ift}}$ because it not only encourages the generation of positive translations like $\mathcal{L}_{\text{ift}}$ does, but also suppresses the generation of negative translations.

## 3.5. Discussion

In this subsection, we conduct detailed experimental analyses to comprehensively evaluate F2STrans.

**A Comparison Between Our List-wise Loss Function $\mathcal{L}_{\text{list}}$ and Preference Learning Loss Functions.** Our style learning can be realized using traditional preference learn-

ing strategies, such as RLHF (Lambert et al., 2022) and PPO (Schulman et al., 2017). Here, we compare our list-wise loss function $\mathcal{L}_{\text{list}}$ in Eq. 4 with Direct Preference Optimization (DPO) (Rafailov et al., 2024), a leading preference learning method known for its simple and effective training process. Since DPO uses only one positive and one negative data, in our experiment, we align the positive sample of DPO with F2STrans, while the negative data is randomly selected from the negative data of F2STrans. Additionally, following our Eq 5, we also compare the model's performance with the instruction-tuning loss function $\mathcal{L}_{\text{ift}}$ as an auxiliary. The results shown in Figure 5 demonstrate that, under both experimental setups, our list-wise loss function $\mathcal{L}_{\text{list}}$ significantly outperforms DPO. Furthermore, including $\mathcal{L}_{\text{ift}}$ as an auxiliary also improves DPO performance.

**LLM Judge of Function-oriented Guidance.** In Figure 6, we provide an in-depth analysis of LLM judge from two perspectives: the number of fine-grained labels $K$ and the comparison with explicit scoring.

▶ *The Number of Fine-grained Labels $K$.* First, compared to binary labels, the advantage of using more fine-grained labels is evident. Performance initially improves with increasing label details, peaking around $K = 5$, and then declines. This suggests that while finer labels enable LLMs to make more accurate judgments about data quality, excessive refinement may cause confusion in the model.

▶ *The Comparison with Explicit Scoring.* Unlike the log-likelihood score that integrates all labels as shown in Eq. 1, here we directly use the labels generated by LLMs, with the number of labels $K$ set to 5. It is observed that explicit scoring performs worse than our method, indicating that the log-likelihood score contains more valuable information than a simple LLM-generated response. Moreover, even with explicit scoring, the results outperform those achieved with binary labels. This further highlights the importance of fine-grained labeling.

**Evaluation from a Style Perspective.** Using our manually annotated target code within our benchmark as references, we assess the stylistic quality of translations gener-

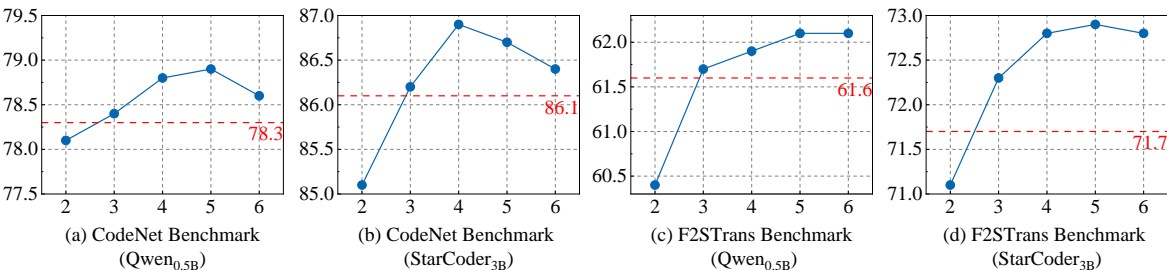

(a) CodeNet Benchmark (Qwen_{0.5B}) (b) CodeNet Benchmark (StarCoder_{3B}) (c) F2STrans Benchmark (Qwen_{0.5B}) (d) F2STrans Benchmark (StarCoder_{3B})

Figure 6: The impact of the fine-grained label count $K$ (blue curve) and the use of explicit scoring (red dashed line) on the results in the LLM judge. In the explicit scoring setting, we set $K$ to 5 and directly use the scores generated by Qwen_{7B} as to measure the relevance between source and target code. In cases of ties among the highest scores, one is selected randomly.

| | **Dis$_{var}$** ↓ | **Dis$_{api}$** ↓ | **Dis$_{stru}$** ↓ | **CSSim** ↑ |
|---|---|---|---|---|
| ● *Qwen$_{32B}$* | | | | |
| Direct | 17.7 | 24.6 | 28.6 | 76.4 |
| CoT | 18.9 | 24.3 | 28.1 | 76.2 |
| RAG | 16.5 | 23.6 | 27.2 | 77.6 |
| Self-debug | 16.3 | 23.4 | 27.0 | 77.8 |
| ● *GPT-4* | | | | |
| Direct | 16.8 | 23.2 | 26.4 | 77.9 |
| CoT | 17.3 | 22.8 | 25.8 | 78.0 |
| RAG | 15.4 | 21.8 | 24.8 | 79.3 |
| Self-debug | 15.2 | 21.8 | 24.5 | 79.5 |
| ● F2STRANS *(Ours)* | | | | |
| Qwen$_{0.5B}$ | 13.8 | 20.1 | 23.8 | 80.7 |
| Qwen$_{1.5B}$ | 12.7 | 19.1 | 22.1 | 82.0 |
| Qwen$_{3B}$ | 11.9 | 18.5 | 20.8 | 82.9 |
| Qwen$_{7B}$ | **11.1** | **17.9** | **19.6** | **83.8** |
| StarCoder$_{3B}$ | 12.0 | 18.3 | 21.1 | 82.8 |

Table 4: Style evaluation of generated translations against the ground truth in our benchmark. The assessment measures code differences in variable naming Dis$_{var}$, API invocation Dis$_{api}$, and code structure Dis$_{stru}$, with CSSim integrating all three aspects. Details of these metrics are shown in Appendix A.

ated by various models, employing the CCSim metric (Li et al., 2024). The results presented in Table 4 demonstrate that F2STRANS significantly improves LLMs' stylistic awareness and generalizes across different scales and types of tested LLMs. A compelling piece of evidence is that F2STRANS-improved Qwen$_{0.5B}$ exhibits superior code style compared to self-debug-based GPT-4, despite GPT-4's better functional performance shown in Table 2. This indicates that functionally correct code can still exhibit stylistic deficiencies. Nonetheless, our F2STRANS effectively mitigates this problem.

**Analysis of F2STRANS in Correcting Errors of Base LLMs.** Based on our benchmark and the previously established code error classification (Pan et al., 2024), we evaluate the effectiveness of function guidance and function-to-style guidance in correcting erroneous translations produced by the naive StarCoder$_{3B}$. As illustrated in Figure 7,

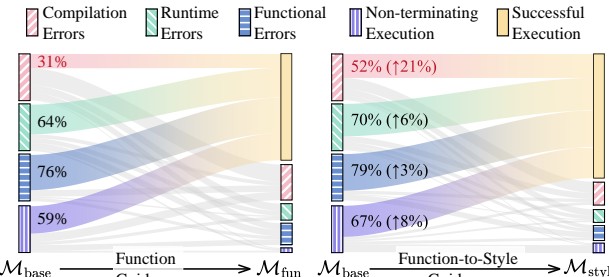

Figure 7: Comparison of error correction in base StarCoder$_{3B}$ translations using function guidance and our function-to-style guidance. For the former, 31% of the compilation errors are successfully corrected, while the latter increases the percentage to 52%.

function guidance successfully rectifies the majority of the base LLMs' translation errors, while only correcting 31% of compilation errors. However, our function-to-style guidance achieves comprehensive improvements across all error types, increasing the correction rate for compilation errors by 21%. This may be because when LLMs diligently adhere to the source code's style, they can avoid superficial code errors such as undeclared identifiers, which often lead to compilation errors. This further underscores the importance of our style guidance.

**Multilingual Modeling Strategies.** We evaluate four multilingual modeling strategies (Yan et al., 2023b) for code translation across the five programming languages used in our work: *one2one* (training separate models for each language pair), *all2one* (translating code from multiple source languages into one target language using a single model), *one2all* (translating code from one source language into multiple target languages using a single model), and *all2all* (a unified model for all language pairs).

As illustrated in Figure 8, the *one2all*, *all2one*, and *all2all* approaches significantly outperform *one2one*, with *all2all* achieving the highest improvements. These results indicate

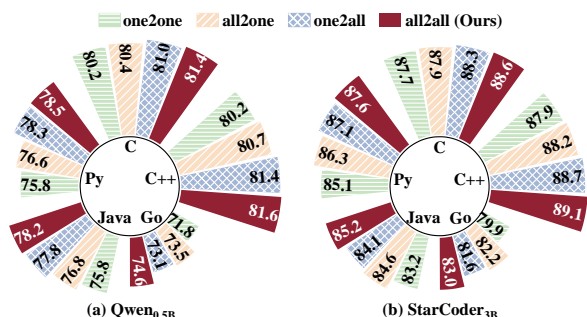

Figure 8: Comparison of various multilingual modeling strategies based on the CodeNet benchmark.

that mixed training across multiple programming languages enhances performance in F2STRANS. Notably, translations from Go exhibit the lowest success rates, which can be attributed to Go's relatively recent emergence in 2009 (Donovan & Kernighan, 2015), resulting in limited knowledge within LLMs. However, the *all2all* strategy provides the most substantial gains when translating from Go, suggesting that it effectively bridges the knowledge gap by enabling LLMs to better understand the relationships between Go and other languages.

## 4. Related Work

Code translation is essential in software development and maintenance, and has been a subject of extensive research for decades (Mossienko, 2003). Early rule-based and program analysis-based methods, such as CxGo (c2g, 2023), were costly and required developers to possess a deep understanding of both source and target languages. In the era of deep learning, Chen et al. (2018) minimizes manual intervention by leveraging a learnable attention mechanism to transform the source language's syntax tree into its counterpart in the target language. To alleviate the reliance on high-quality parallel code pairs, a series of unsupervised training strategies have been proposed (Xue et al., 2024; Rozière et al., 2022). For example, Lachaux et al. (2020) employs masked pre-training on large-scale monolingual code data and back-translation strategies to learn the mapping between source and target languages. Another approach is to use code solutions to identical programming problems in different languages from competitive platforms as weakly supervised parallel data (Ahmad et al., 2021; Zhu et al., 2022; Yan et al., 2023a; Xie et al., 2023).

Recent advancements in LLM-based code translation have been substantial. Yan et al. (2023b) evaluated ChatGPT's performance on code translation tasks using standard inference techniques such as direct prompting, few-shot learning, and CoT, demonstrating the effectiveness of LLMs. Researchers have since sought to enhance LLMs' code translation capabilities from multiple angles. Macedo et al. (2024)

explored the impact of output formatting on LLM performance. Bhattarai et al. (2024a;b) improved translations of low-resource programming languages using a RAG strategy. Yang et al. (2024); Pan et al. (2024); Yin et al. (2024) introduced the self-debugging strategy, where LLM-generated target code is compiled, and any detected bugs are incorporated into subsequent prompts to guide precise fixes. Huang et al. (2023); Szafraniec et al. (2022); Sun et al. (2024) used a unified intermediate representation as a pivot for translating between programming languages, enabling models to capture language-agnostic code semantics effectively. Additionally, multi-agent systems (Yuan et al., 2024) and human-machine interactive systems (Liu et al., 2024b) leveraging LLMs provide developers with more transparent reasoning processes and facilitates the generation of higher-quality translated code.

## 5. Conclusion

In this study, we introduced a novel feature-to-style training paradigm to effectively adapt LLMs for code translation tasks. Initially, we conducted functional learning using high-quality source–target code pairs from online programming platforms, generating functionally correct translations. Subsequently, we applied style learning based on positive and negative translation data, yielding more readable and stylistically consistent translations. This two-stage training paradigm significantly improved the performance of LLMs across various scales and types on our newly constructed benchmark as well as traditional code translation benchmarks, even surpassing GPT-4. The substantial gains achieved by our approach highlight its effectiveness in advancing code translation capabilities of LLMs.

## Impact Statement

This work introduces a function-to-style guiding paradigm to improve the correctness and readability of code generated by translation models. We anticipate that our approach will contribute to more efficient software development and maintenance. However, it is crucial to acknowledge the potential for misuse, such as the translation of malicious code. Ultimately, adhering to ethical guidelines is essential to ensure the responsible application of this technology.

## Acknowledgement

This work was supported in part by National Science Foundation of China (62336008, 62476070, 62125201, U24B20174), Shenzhen Science and Technology Program (JCYJ20241202123503005, GXWD20231128103232001, ZDSYS20230626091203008, KQTD2024072910215406) and Department of Science and Technology of Guangdong (2024A1515011540).

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

## A. Background

**CCSim.** CCSim, proposed by Li et al. (2024), measures stylistic similarity of code by considering the edit distances (Ristad & Yianilos, 1998) of variable naming, API invocation, and code structure.

▶ *Calculation of Variable Name Edit Distance* — $\mathrm{Dis}_{\mathrm{var}}$. First, extract all variable names, $V_1$ and $V_2$ from the two code respectively. Then, compute the edit distance between these two sets of variables as follows:

$$\mathrm{Dis}_{\mathrm{V}_1} = \frac{1}{||\lambda||_1} \sum_{v_i \in \mathrm{V}_1} \lambda_i \min_{v_j \in \mathrm{V}_2} \mathrm{ED}(v_i, v_j)$$

$$\mathrm{Dis}_{\mathrm{V}_2} = \frac{1}{||\lambda||_1} \sum_{v_i \in \mathrm{V}_2} \lambda_i \min_{v_j \in \mathrm{V}_1} \mathrm{ED}(v_i, v_j) \tag{6}$$

$$\mathrm{Dis}_{\mathrm{var}} = \frac{\mathrm{Dis}_{\mathrm{V}_1} + \mathrm{Dis}_{\mathrm{V}_2}}{2},$$

where ED is the Edit Distance (Ristad & Yianilos, 1998), and $\lambda_i$ is the normalized inverse document frequency (IDF) (Sparck Jones, 1972) of a variable naming $v_i$, which is used to decrease the impact of common words.

▶ *Calculation of API Invocation Edit Distance* — $\mathrm{Dis}_{\mathrm{api}}$. It is calculated similarly to $\mathrm{Dis}_{\mathrm{var}}$, except that variable names are replaced with API names.

▶ *Calculation of Code Structure Edit Distance* — $\mathrm{Dis}_{\mathrm{stru}}$. The measurement of code structure is based on the Tree Edit Distance (TED) (Paaßen, 2018) of abstract syntax tree. Specifically, it measures the structural difference of two code snippets by determining the fewest insertions, deletions, and replacements needed to transform one tree into the other.

Based on the edit distances of variable names, API invocation, and code structure described above, CCSim measures the stylistic similarity between two code snippets as follows:

$$\mathrm{CSSim} = 1 - \frac{\mathrm{Dis}_{\mathrm{var}} + \mathrm{Dis}_{\mathrm{api}} + \mathrm{Dis}_{\mathrm{stru}}}{3}, \tag{7}$$

where $\mathrm{CSSim}, \mathrm{Dis}_{\mathrm{var}}, \mathrm{Dis}_{\mathrm{api}}, \mathrm{Dis}_{\mathrm{stru}} \in [0, 1]$, and higher CCSim values indicate greater similarity.

**Computational Accuracy.** We utilize Computational Accuracy (CA) to assess the functional correctness of code translations produced by various models. Given all source code, their corresponding target code generated by the models, and the input data $\{(src_1, tgt_1, \mathrm{INPUT}_1), \ldots, (src_N, tgt_N, \mathrm{INPUT}_N)\}$, CA is calculated as follows:

$$\mathrm{CA} = \frac{\sum_{k=1}^{N} \mathrm{ca}\left(src_k, \hat{tgt_k}\right)}{N}$$

$$\mathrm{ca}\left(src_k, \hat{tgt_k}\right) = \begin{cases} 1, & \text{if } \mathrm{exec}\left(src_k, input\right) = \mathrm{exec}\left(tgt_k, input\right), \forall input \in \mathrm{INPUT}_k \\ 0, & \text{otherwise} \end{cases} \tag{8}$$

where $\mathrm{exec}(.)$ denotes the result of executing the code with a given input.

## B. More Implementation Details.

In the function-oriented training, we construct approximately 5,000 code pairs for each translation scenario, such as translating from C++ to Python, with a corresponding scale of 10,000 in the style-oriented training. Throughout both training stages, we maintain consistent hyperparameters, employing 2 epochs and a learning rate of $1 \times 10^{-5}$. During inference, we set the temperature of the LLMs to 0.7. All our experiments are carried out on a machine equipped with eight NVIDIA A800-SXM4-80GB GPUs.

## C. Baseline Details

We adopt $\mathrm{Qwen}_{32B}$ and GPT-4 as our baselines, implementing four previously established prompt learning strategies. The prompts are shown in Appendix F.2.

- **Direct prompt learning (Direct).** This straightforward strategy (Yang et al., 2024) provides the LLM with a prompt that includes the source language, target language, source code, and a concise task description. Owing to the powerful

| Method | LLM | Translation C → {} | | | | Translation C++ → {} | | | | Translation Go → {} | | | | Translation Java → {} | | | | Translation Py → {} | | | | Avg. |
|---|---|---|---|---|---|---|---|---|---|---|---|---|---|---|---|---|---|---|---|---|---|---|
| | | C++ | Go | Java | Py | C | Go | Java | Py | C | C++ | Java | Py | C | C++ | Go | Py | C | C++ | Go | Java | |
| Direct | Qwen$_{32B}$ | 90.3 | 57.1 | 81.6 | 62.5 | 73.6 | 54.6 | 77.0 | 56.8 | 79.2 | 83.1 | 80.4 | 61.7 | 78.1 | 75.3 | 49.9 | 68.6 | 70.0 | 71.0 | 61.5 | 78.5 | 70.5 |
| CoT | | 85.9 | 61.9 | 76.6 | 63.1 | 74.5 | 51.9 | 76.3 | 62.2 | 74.7 | 83.4 | 81.2 | 66.3 | 75.7 | 77.0 | 45.0 | 70.7 | 68.4 | 75.6 | 65.3 | 82.6 | 70.9 |
| RAG | | 87.3 | 69.5 | 78.7 | 67.9 | 77.7 | 65.1 | 74.3 | 58.2 | 83.5 | 82.2 | 78.5 | 68.1 | 78.0 | 77.1 | 60.8 | 74.5 | 74.5 | 76.1 | 69.0 | 83.0 | 74.2 |
| Self-debug | | 91.2 | 64.9 | 83.2 | 67.4 | 82.5 | 58.6 | 78.1 | 64.4 | 82.2 | 85.1 | 82.6 | 68.2 | 79.4 | 82.5 | 55.8 | 77.3 | 73.6 | 79.7 | 69.8 | 84.3 | 75.5 |
| Direct | GPT-4 | 95.5 | 64.5 | 90.2 | 65.6 | 87.9 | 63.5 | 85.5 | 66.6 | 87.5 | 91.4 | 88.1 | 77.4 | 84.1 | 82.2 | 57.4 | 71.9 | 71.5 | 76.0 | 58.9 | 76.6 | 77.1 |
| CoT | | 92.6 | 70.7 | 86.5 | 65.3 | 83.7 | 62.8 | 85.7 | 71.0 | 85.3 | 92.6 | 88.6 | 80.6 | 84.2 | 85.4 | 54.5 | 72.9 | 70.6 | 77.7 | 64.8 | 83.8 | 78.0 |
| RAG | | 93.5 | 72.1 | 91.3 | 70.1 | 84.4 | 67.9 | 83.9 | 74.1 | 85.6 | 91.5 | 85.9 | 84.3 | 85.1 | 84.9 | 64.2 | 76.2 | 77.1 | 82.9 | 66.8 | 84.7 | 80.3 |
| Self-debug | | 96.9 | 70.2 | 90.3 | 75.0 | 89.2 | 68.1 | 87.1 | 75.1 | 88.9 | 93.4 | 89.1 | 84.7 | 85.6 | 90.2 | 64.6 | 82.3 | 78.3 | 81.3 | 65.1 | 85.0 | 82.0 |
| **F2STRANS (Ours)** | Qwen$_{0.5B}$ | 96.2 | 76.0 | 87.5 | 77.3 | 87.0 | 73.6 | 80.3 | 72.1 | 80.5 | 84.7 | 84.2 | 81.4 | 78.9 | 81.2 | 68.9 | 80.7 | 64.9 | 70.0 | 60.6 | 71.9 | 77.9 |
| | Qwen$_{1.5B}$ | 97.3 | 83.8 | 91.6 | 82.7 | 90.2 | 82.8 | 87.1 | 78.9 | 87.2 | 90.3 | 90.1 | 87.6 | 85.6 | 86.0 | 79.0 | 90.3 | 76.3 | 80.0 | 73.7 | 85.4 | 85.3 |
| | Qwen$_{3B}$ | 97.3 | 86.6 | 93.9 | 83.7 | 91.9 | 85.2 | 89.6 | 81.2 | 88.2 | 92.1 | 91.4 | 89.5 | 86.8 | 89.4 | 81.6 | 90.8 | 79.2 | 84.3 | 77.4 | 88.0 | 87.4 |
| | Qwen$_{7B}$ | 97.7 | 88.4 | 94.6 | 84.9 | 93.0 | 86.8 | 89.9 | 85.4 | 89.5 | 93.6 | 93.2 | 91.0 | 89.8 | 90.4 | 83.7 | 93.5 | 81.2 | 85.8 | 80.1 | 89.3 | 89.1 |
| | StarCoder$_{3B}$ | 97.5 | 86.1 | 92.4 | 84.8 | 91.2 | 84.9 | 88.9 | 82.1 | 88.8 | 91.6 | 90.9 | 88.1 | 86.0 | 89.1 | 80.6 | 90.8 | 78.0 | 83.0 | 76.8 | 85.9 | 86.9 |

Table 5: Code translation results of various models on XcodeEval benchmarks.

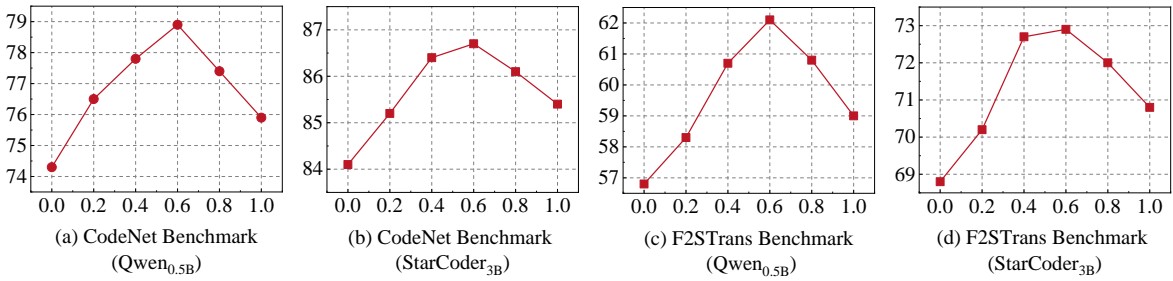

| (a) CodeNet Benchmark | (b) CodeNet Benchmark | (c) F2STrans Benchmark | (d) F2STrans Benchmark |
|---|---|---|---|
| (Qwen$_{0.5B}$) | (StarCoder$_{3B}$) | (Qwen$_{0.5B}$) | (StarCoder$_{3B}$) |

Figure 9: The impact of trade-off hyper-parameters $\beta$ in the loss function Eq. 5 on the results.

instruction-following abilities of LLMs, direct prompt learning has proven highly effective across a broad range of tasks (Du et al., 2024; Shi & Zhou, 2023).

- **Chain of thought (CoT).** This strategy, proposed by Yan et al. (2023b), first encourages the model to thoughtfully consider the translation process and identify potential challenges before undertaking code translation. This strategy usually achieves better results than direct prompt learning (Lee et al., 2024).
- **Retrieval-augmented generation (RAG).** Consistent with previous RAG-based code translation approaches (Bhattarai et al., 2024a;b), we retrieve the most similar source-target pair from our style-oriented positive translation dataset using the code embedding model Jina. This retrieved pair provides auxiliary context for the model.
- **Self-debug prompt learning (Self-debug).** This method (Yang et al., 2024; Pan et al., 2024) employs LLMs to generate initial translations and test cases, verifies the translated code's correctness using these tests, and subsequently corrects any errors based on the test results and compiler error messages.

# D. Additional Results

We further evaluate F2STRANS on xCodeEval (Khan et al., 2024), as shown in Table 5. We can find that F2STRANS continues to demonstrate a significant advantage. Notably, F2STRANS enables Qwen$_{0.5B}$ to outperform both Qwen$_{32B}$ and GPT-4 on average across 20 code translation tasks. Qwen$_{1.5B}$ even surpasses the self-debugging GPT-4.

# E. Additional Discussion

**The $\beta$ in the Loss Function Eq. 5 of Style Learning.** Figure 9 illustrates the performance of the model with different trade-off hyperparameters $\beta$. It is evident that, during the style learning, as the weight of the instruction fine-tuning loss $\mathcal{L}_{\text{list}}$ increases, the model's performance improves. The best performance is achieved when $\beta \approx 0.6$, after which the performance gradually declines. Furthermore, when $\beta = 1.0$ (i.e., using only $\mathcal{L}_{\text{list}}$), the model consistently outperforms the case when $\beta = 0.0$ (i.e., using only $\mathcal{L}_{\text{ift}}$). This is because $\mathcal{L}_{\text{list}}$ takes both positive and negative data into account, whereas $\mathcal{L}_{\text{ift}}$ considers only the positive data.

# F. Prompt Settings

## F.1. Prompts Used by F2STRANS

---
**LLM-judge Prompt.**

You are given a source code in {SOURCE_LANG} and a translated code in {TARGET_LANG}. Please evaluate the translation by scoring it on a scale from 1 to 5, where:
1: The translated code has significant differences in logic, structure, or implementation compared to the source code, and would likely not work as intended.
2: The translated code works, but there are noticeable differences in logic, style, or structure that deviate from the original solution.
3: The translated code is mostly similar to the source code but has minor differences or optimizations that do not impact overall functionality.
4: The translated code is very close to the original code, with minor, non-critical deviations in style or structure.
5: The translated code is highly consistent with the source code, both in terms of logic and structure, and works as intended.
### {SOURCE_LANG} Code:
{SOURCE_CODE}
### {TARGET_LANG} Code:
{TARGET_CODE}
### Score:

---

---
**Style-aware Prompt.**

Translate the following {SOURCE_LANG} code to {TARGET_LANG} while preserving the source code style, including variable names, function names, and code structure. Adhere to the following guidelines:
1. Variable and Function Names:
- Maintain the same variable and function names as in the source code.
- If necessary due to language constraints, adjust names minimally while keeping them similar to the original.
2. Code Structure:
- Preserve the overall structure and logic flow of the source code.
- Maintain the same control structures (e.g., loops, conditionals) and their nesting levels.
3. Libraries and APIs:
- Replace source language libraries and functions with equivalent {TARGET_LANG} libraries and functions.
- Keep the variable and parameter names the same as in the source code where possible.
4. Comments:
- Retain any comments present in the source code.
- Translate comments to {TARGET_LANG} if applicable, maintaining their position and style.
5. Code Formatting:
- Maintain a similar code formatting style to the source code, including indentation, spacing, and line breaks, as much as possible within the conventions of {TARGET_LANG}.
Print only the translated {TARGET_LANG} code and end with the comment "End of Code".
### Source code:
{SOURCE_CODE}

---

---
**Prompts $\mathcal{P}$ in Eq. 2 and 4 During the Training and Inference of F2STRANS.**

Translate the {SOURCE_LANG} code to {TARGET_LANG} code.
### {SOURCE_LANG} Code:
{SOURCE_CODE}
### {TARGET_LANG} Code:

---

## F.2. Prompts Used by Baselines

**Direct Prompt Learning.**

Translate the following {SOURCE_LANG} code to {TARGET_LANG}. Print only the {TARGET_LANG} code and end with the comment "End of Code".
### Source code:
{SOURCE_CODE}

**Chain-of-thought Prompt Learning.**

First, understand the functionality of the following {SOURCE_LANG} code and predict the execution output. Then, translate the {SOURCE_LANG} code into {TARGET_LANG} while maintaining the same functionality, ensuring that the translated code can be successfully executed.
### Source code:
{SOURCE_CODE}

**Retrieval Augmented Generation Prompt Learning.**

Translate the following {SOURCE_LANG} code to {TARGET_LANG}. Print only the {TARGET_LANG} code and end with the comment "End of Code".
### Source code:
{SOURCE_CODE_EXAMPLE}
### Target code:
{TARGET_CODE_EXAMPLE}
### Source code:
{SOURCE_CODE}
### Target code:

**Self-debug Prompt Learning When Effect is COMPILE ERROR or RUNTIME ERROR.**

You were asked to translate the following {SOURCE_LANG} code to {TARGET_LANG}:
{SOURCE_CODE}
Your response was the following {TARGET_LANG} code:
{TRANSLATED_CODE}
Executing your generated code gives the following error because it is syntactically incorrect:
{STDERR}
Can you re-generate your response and translate the above {SOURCE_LANG} code to {TARGET_LANG}. Print only the {TARGET_LANG} code and do not add any other natural language description in your output. Make sure your generated code is syntactically correct.

**Self-debug Prompt Learning When Effect is INCORRECT OUTPUT.**

You were asked to translate the following {SOURCE_LANG} code to {TARGET_LANG}:
{SOURCE_CODE}
Your response was the following {TARGET_LANG} code:
{TRANSLATED_CODE}
Executing your generated code gives the following output:
{GENERATED_OUTPUT}
instead of the following expected output:
{EXPECTED_OUTPUT}
Can you re-generate your response and translate the above {SOURCE_LANG} code to {TARGET_LANG}. Print only the {TARGET_LANG} code and do not add any other natural language description in your output. Make sure your generated code is syntactically correct. Your generated {TARGET_LANG} code should take the following input and generate the expected output:
Input:
{TEST_INPUT}
Expected Output:
{EXPECTED_OUTPUT}

