# OpenReview forum: "Function-to-Style Guidance of LLMs for Code Translation"
_ICML.cc/2025/Conference — ICML 2025 poster_

### Official Review · Reviewer_stzs · 2025-03-12

**Overall Recommendation:** 4

**Summary:**

This paper studies code translation. Different from previous settings where researchers only care about the functional correctness of translated codes, the authors in this paper also care the functional consistency. They propose a method to handle this by splitting it into a two-stage training framework, function learning and style learning. Besides existing benchmarks, the authors also propose a more up-to-date benchmark for code translation. Experimental results demonstrate the effectiveness of their proposed ;earning framework and extensive ablation studies explore the impact of different design choices.

**Claims And Evidence:**

Yes. Experimental results demonstrate that style learning benefits code translation.

**Essential References Not Discussed:**

N.A

**Experimental Designs Or Analyses:**

There are other benchmarks that could be used for code translation, other than CodeNet, though not specifically designed for code translation. For example, the Multiple-E [1]. The author should also apply their framework on such benchmarks.

[1] Multi-Programming Language Evaluation of Large Language Models of Code

**Methods And Evaluation Criteria:**

In style learning, the authors use a list-wise loss function to train LLMs to favor style-consistent target codes. The motivation of using such a loss function is not clear. This preference could also be learned through reinforcement learning (e.g DPO). The authors should study the impact of replacing the list-wise loss function with RL algorithms in their ablation studies.

At least for me, the performance of list-wise loss doe not have to be better than DPO. This is because I consider these loss functions as designing variations of the framework. In the future, there may be more advanced loss functions to achieve even better performance.

**Other Comments Or Suggestions:**

N.A

**Other Strengths And Weaknesses:**

N.A

**Questions For Authors:**

See my comments above.

**Relation To Broader Scientific Literature:**

This work is specifically designed for code translation.

**Theoretical Claims:**

There are no theoretical claims in this paper.

---

> ### Author Rebuttal · Authors · 2025-04-01
>
> We greatly appreciate the reviewer's insightful and constructive feedback, and we have carefully addressed each point in our response to resolve your concerns.
> If our response has satisfactorily addressed your questions, we kindly request your consideration of raising the score (currently Rating: 3: Weak Accept).
> Should any further issues remain, please feel free to share your additional comments, and we will continue actively responding to your comments and improving our submission.
>
> >**Q1:**
> Comparison between DPO and list-wise loss function.
>
> **A1:**
> The list-wise loss function is designed to improve the generation of positive translations by learning from one positive example and multiple negative examples.
> This approach differs significantly from DPO, which optimizes models based on pairwise comparisons between a single positive and a single negative sample.
>
> To evaluate the effectiveness of these two methods, we conduct experiments on the CodeNet and F2STrans benchmarks, with the results presented in **Figure 8 of Appendix E** in our paper.
> The average performance across three training runs is summarized in the following table, from which we can observe that our list-wise loss function is more effective.
> For more details on the experimental setup and additional analyses, please refer to **Figure 8 of Appendix E** in our paper.
>
> |Dataset|Our Models|DPO|List-wise Loss|Delta|
> |-|-|-|-|-|
> |CodeNet|Qwen-0.5B|78.2|78.9|+0.7|
> ||StarCoder-3B|85.5|86.7|+1.2|
> |F2STrans|Qwen-0.5B|61.1|62.1|+1.0|
> ||StarCoder-3B|71.4|72.9|+1.5|
>
> ---
>
> >**Q2:**
> In addition to the CodeNet and F2STrans benchmarks, the paper should evaluate performance on a broader range of benchmarks, such as Multiple-E.
>
> **A2:**
> We fully recognize the importance of comprehensive testing benchmarks.
> Therefore, **Table 5 of Appendix D** in our paper demonstrates the model's performance on the **xCodeEval** and **CodeScope** benchmarks.
> The average translation results for various source languages ​​are as follows:
>
> |xCodeEval|C|C++|Go|Java|Py|Avg.|
> |-|-|-|-|-|-|-|
> |Direct (Qwen-32B)|72.9|65.5|76.1|68.0|70.3|70.5|
> |CoT (Qwen-32B)|71.9|66.2|76.4|67.1|73.0|70.9|
> |RAG (Qwen-32B)|75.8|68.8|78.1|72.6|75.6|74.2|
> |Self-debug (Qwen-32B)|76.7|70.9|79.5|73.8|76.9|75.5|
> |Direct (GPT4)|79.0|75.9|86.1|73.9|70.8|77.1|
> |CoT (GPT4)|78.8|75.8|86.8|74.2|74.3|78.0|
> |RAG (GPT4)|81.8|77.6|86.8|77.6|77.9|80.3|
> |Self-debug (GPT4)|83.1|79.9|89.0|80.7|77.4|82.0|
> |Ours (Qwen-0.5B)|84.2|78.2|82.7|77.4|66.8|77.9|
> |Ours (Qwen-1.5B)|88.8|84.8|88.8|85.2|78.8|85.3|
> |Ours (Qwen-3B)|90.4|87.0|90.3|87.1|82.2|87.4|
> |Ours (Qwen-7B)|**91.4**|**88.8**|**91.8**|**89.3**|**84.1**|**89.1**|
> |Ours (StarCoder-3B)|90.2|86.8|89.8|86.6|80.9|86.9|
>
>
> |CodeScope|C|C++|Go|Java|Py|Avg.
> |-|-|-|-|-|-|-
> |Direct (Qwen-32B)|59.4|56.0|67.4|60.7|48.6|58.4
> |CoT (Qwen-32B)|67.6|53.5|65.3|60.1|52.8|59.9
> |RAG (Qwen-32B)|75.4|58.8|65.7|67.8|59.7|65.5
> |Self-debug (Qwen-32B)|75.3|61.4|71.8|67.9|60.7|67.4
> |Direct (GPT4)|69.3|64.4|77.8|62.1|49.3|64.6
> |CoT (GPT4)|74.6|65.6|75.9|68.2|55.0|67.9
> |RAG (GPT4)|80.1|68.2|77.4|73.5|59.8|71.8
> |Self-debug (GPT4)|80.8|70.5|81.8|74.8|61.2|73.8
> |Ours (Qwen-0.5B)|75.4|69.4|75.8|67.9|45.7|66.8
> |Ours (Qwen-1.5B)|84.5|71.9|78.2|76.8|58.2|73.9
> |Ours (Qwen-3B)|88.6|75.3|80.7|82.5|65.5|78.5
> |Ours (Qwen-7B)|**92.8**|**86.6**|82.6|**85.3**|**68.5**|**83.2**|
> |Ours (StarCoder-3B)|87.3|78.1|**83.2**|78.9|62.5|78.0
>
>
> In response to your suggestion, we have conducted additional tests on the **Multiple-E** and **CodeTransOcean** benchmarks.
> For Multiple-E, since it does not include C code, we focus on evaluating its performance across other languages.
> For CodeTransOcean, we adopt the CodeBLEU metric, consistent with the standard practice described in the CodeTransOcean paper.
> For other benchmarks, we continue to use the Computational Accuracy metric as outlined in our paper.
>
> |Multiple-E|C++|Go|Java|Py|Avg.
> |-|-|-|-|-|-
> |Direct (Qwen-32B)|58.6|64.8|72.0|44.3|59.9
> |CoT (Qwen-32B)|61.8|65.8|77.0|47.3|63.0
> |RAG (Qwen-32B)|62.9|69.4|77.5|49.1|64.7
> |Direct (GPT4)|71.2|75.9|78.4|46.5|68.0
> |CoT (GPT4)|71.6|77.6|79.1|48.8|69.3
> |Self-debug (GPT4)|73.5|77.2|81.6|52.3|71.1
> |Ours (Qwen-0.5B)|71.7|71.9|81.7|40.0|66.3
> |Ours (Qwen-1.5B)|79.3|77.9|89.5|50.0|74.2
> |Ours (Qwen-3B)|82.6|80.6|92.4|54.7|77.6
> |Ours (Qwen-7B)|**84.6**|**83.6**|**93.3**|**58.2**|**79.9**|
> |Ours (StarCoder-3B)|82.3|80.9|92.1|52.9|77.1
>
> |CodeTransOcean|C|C++|Go|Java|Py|Avg.
> |-|-|-|-|-|-|-
> |Direct (Qwen-32B)|27.6|27.0|27.1|31.6|24.1|27.5
> |CoT (Qwen-32B)|27.3|26.5|27.5|32.1|25.2|27.7
> |RAG (Qwen-32B)|28.2|27.4|28.1|32.4|24.7|28.2
> |Direct (GPT4)|29.1|29.0|28.4|32.9|25.7|29.0
> |CoT (GPT4)|29.2|29.3|28.6|32.2|26.2|29.1
> |Ours (Qwen-0.5B)|30.3|29.0|28.4|30.8|25.7|28.9
> |Ours (Qwen-1.5B)|30.8|29.7|29.8|32.3|25.8|29.7
> |Ours (Qwen-3B)|**31.2**|29.9|**30.9**|33.2|26.7|30.4
> |Ours (Qwen-7B)|**31.2**|**30.6**|30.4|**34.3**|**27.1**|**30.7**
> |Ours (StarCoder-3B)|30.8|29.8|30.0|32.8|26.6|30.0

---

> > ### Comment · Reviewer_stzs · 2025-04-03
> >
> > I thank the authors for their rebuttal. All my concerns are well addressed. Thus, I will raise my score to 4. Please make sure that all of the content during the rebuttal will appear in the camera-ready version.

---

> > > ### Author Response · Authors · 2025-04-03
> > >
> > > Thank you very much for your thoughtful feedback and for raising your score to 4! We deeply appreciate the time and effort you have invested in carefully reviewing our work and providing such constructive suggestions. Your insights have been invaluable in helping us strengthen our paper and address key concerns more comprehensively.
> > >
> > > We will make sure to incorporate the additional results and discussions into the camera-ready version, as you suggested. Your feedback has not only improved the quality of our work but also deepened our understanding of the methods and their implications.
> > >
> > > Once again, thank you for your support and for recognizing our efforts. We hope that our work can contribute meaningfully to the field.

---

### Official Review · Reviewer_a86m · 2025-03-13

**Overall Recommendation:** 4

**Summary:**

This paper proposes F2STrans, an approach for improving code-to-code translation by optimizing large language models (LLMs). The main idea of this paper is to provide fine-tuning processes that instruct knowledge on good code and bad code in terms of two criteria: Functional Learning and Style Learning. For functional learning, which requires LLM to generate translated code with a high level of functional correctness, this approach designs a paradigm called Function-oriented Guidance by two steps: Function-consistent Data Construction using code embedding and ranking candidate code using LLMs and Functional Learning, which uses Instruction Fine-tuning to return the optimized LLM for code translation. For Style learning, this approach uses strong LLMs to generate positive translation examples with stylistic similarity scores to find the best example and the same process to find negative translation examples. Next, it uses a lit-wise loss function to instruct LLM to generate code with better style. In the experiment, this work uses CodeNet and their proposed benchmark called F2STrans for evaluation. Their results show that F2STrans can provide better Computational Accuracy and stylistic accuracy compared to state-of-the-art LLMs like GPT-4 and Qwen-32B.

**Claims And Evidence:**

The claims made in the submission are clear and convincing.

**Essential References Not Discussed:**

I suggest the authors explain why they chose stylistic and functional correctness as two aspects for improvement. They should add papers about studies on code preferences (CodeUltraFeedback: An LLM-as-a-Judge Dataset for Aligning Large Language Models to Coding Preferences - Wessay et al.) to describe the potential/ challenges of applying new aspects to their approach.

**Experimental Designs Or Analyses:**

Experiment designs and analysis are sound and valid.

**Methods And Evaluation Criteria:**

The evaluation criteria are clear and convincing, with the benchmark constructed and the evaluation metrics. For the proposed method, It is good to know the correctness of Judge LLM used in Function Learning and how it can impact the accuracy of code translation. In Table 3, the authors reported the accuracy without LLM Judge and it underperforming the default configuration. However, the authors need to analyze the case when the judge returns bad results. Other parts of the Methods are clear and convincing.

**Other Comments Or Suggestions:**

Lack of comparison with non-LLM approaches. Authors can consider comparing their work with transformer-based approaches such as CodeRosetta (CodeRosetta: Pushing the Boundaries of Unsupervised Code Translation for Parallel Programming).

**Other Strengths And Weaknesses:**

Strength:
- This paper is well-written and has a good structure to follow.
- The strategy of providing better samples for the fine-tuning process is a good direction.
- The selections of loss functions for functional learning and style-oriented learning were based on well-known prior works.
- The result has an acceptable level of outperformance compared to existing works.
- Integrating manually annotated ground-truth translation.

Weakness:
- Unclear motivation example: Can you define which part of the translated code should include the code for I/O optimization?
- For Function-consistent Data Construction, there is a risk that the LLM Judge can provide incorrect output. Further study is needed to analyze the case when LLM judges provide low-quality output.
- While human annotation for ground-truth dataset creation is important, there is are lack of description of this process in the current version of this paper.

**Questions For Authors:**

1. The scale of the manual ground-truth translation creation  (mentioned in section 3.1). How many people are required to do this process? How do authors ensure that human responses were correct?

2.Code Pair embedding. Did authors try other well-known code embedding methods, such as UniXcoder (Guo et al. - UniXcoder: Unified Cross-Modal Pre-training for Code Representation)?

**Relation To Broader Scientific Literature:**

This paper proposed a new direction of improving LLM-based code translation with a focus on important aspects of translated code as functional correctness and stylistic information. Their results show that with proper improvement, simple open LLMs can still outperform large scale open LLM and closed LLMs. They also contributed to the science community a dataset that will be helpful in evaluating code translation.

**Theoretical Claims:**

This paper proofs that the existing dataset for evaluating LLM-based code translation like CodeNet is somehow outdated. They claim that their proposed dataset can capture better the complexity and nature of LLM-based code translation. Through the experiment, they successfully proved that claim since the accuracy of multiple configurations of their tool for code translation tends to be lower with their constructed benchmark.

---

> ### Author Rebuttal · Authors · 2025-04-01
>
> We thank the reviewer for the insightful and valuable comments.
>
> >**Q1:**
> Further study is needed to analyze the case when LLM judges provide low-quality output.
>
> **A1:**
> We observe that LLM Judge is prone to errors when evaluating source code that is either lengthy or contains numerous built-in APIs, e.g., "map", "lambda", etc., in Python.
>
> To evaluate its performance in assessing translation quality, we introduce an **LLM Judge benchmark**.
> This benchmark comprises source code and human-annotated translation from our F2STrans benchmark, along with translations generated by Qwen-32B and GPT-3.5.
> We measure the performance of a Qwen-7B-based LLM Judge by its success rate in identifying the human-annotated translation as the best translation for a given source code.
> The table below summarizes results for source code with varying API counts and token lengths.
>
> While performance deteriorates with increased complexity, our analysis indicates that LLM Judge still reliably identifies high-quality translations in most scenarios.
>
> #API|[0,3]|[4,7]|[8,11]|Overall
> -|-|-|-|-
> Results|83.4|75.6|61.3|77.5
>
> #Token|[0,250]|[251,500]|[501,750]|[750, $+\infty$]|Overall
> -|-|-|-|-|-
> Results|84.3|79.6|64.6|46.3|77.5
>
> ---
>
> >**Q2:**
> Why were style and functional correctness chosen as two aspects for improvement?
>
> **A2:**
> We focused on style and functional correctness for two main reasons:
> 1. **Fundamental requirements**: Functional correctness ensures the translated code maintains the same behavior as the source code—an essential requirement for any translation system.
> Stylistic correctness enhances readability and maintainability, making the translated code easier to understand and modify.
> 2. **Translation-specific factors**: While efficiency, security, and other aspects are certainly important in code development, these factors are largely determined by the source code itself rather than the translation process.
>
> We are actively exploring ways to optimize code translation quality from a broader perspective in our ongoing research.
>
> ---
>
> >**Q3:**
> The research on code preferences (CodeUltraFeedback) should be discussed.
>
> **A3:**
> We agree that the CodeUltraFeedback is highly relevant to our work.
> It inspires us to explore incorporating user-defined code styles and preferences into our code translation system.
> Additionally, it illuminates important future challenges beyond our current scope, particularly how to extend our approach to address other crucial non-functional factors like efficiency, security, and maintainability in the translated code.
> Our paper will incorporate a more detailed discussion of CodeUltraFeedback and its implications for our research.
>
> ---
>
> >**Q4:**
> Clarification on I/O optimization in motivation example.
>
> **A4:**
> In the C++ code, the lines
> ```cpp
> ios::sync_with_stdio(false);
> cin.tie(0);
> ```
> disable synchronization between C++ I/O streams and untie cin from cout, reducing overhead and accelerating input/output operations.
> In Python, a comparable optimization would involve using faster input methods, such as ```sys.stdin.readline()``` or reading all input at once via ```sys.stdin.read()```.
>
> ---
>
> >**Q5:**
> The paper lacks a detailed description of the process for manually annotating the ground-truth dataset.
>
> **A5:**
> Our annotation team consisted of 30 professional software developers with industry experience, 15 graduate students, and 5 doctoral students in computer science.
> During code translation, each source code was translated by at least 5 different annotators, ensuring multiple candidate translations.
> Doctoral students then selected the best translation or refined problematic ones.
> For test case annotation, we maintained a 95% branch coverage rate, prioritizing edge cases and challenging scenarios.
> The entire annotation process took approximately 5 months.
>
> ---
>
> >**Q6:**
> Lack of comparison with non-LLM approaches, like CodeRosetta.
>
> **A6:**
> We appreciate the suggestion to compare our work with non-LLM approaches for a more comprehensive evaluation.
> While CodeRosetta focuses on translations between C++ and CUDA/Fortran (which differs from our scope), we will include a more detailed discussion of it in our paper.
> Below, we compare our approach with two prominent non-LLM code translation systems, TransCode-ST and TransCode-IR.
>
> Codenet Benchmark|C++↔Go|C++↔Java|C++↔Py|Go↔Java|Java↔Py
> -|-|-|-|-|-
> TransCoder-ST|-|66.3|72.7|-|71.2
> TransCoder-IR|54.3|69.4|-|50.6|-
> Ours (Qwen-0.5B)|77.9|84.9|78.2|77.3|80.6
>
>
> F2STrans Benchmark|C++↔Go|C++↔Java|C++↔Py|Go↔Java|Java↔Py
> -|-|-|-|-|-
> TransCoder-ST|-|50.2|26.3|-|32.6
> TransCoder-IR|41.7|53.2|-|54.3|-
> Ours (Qwen-0.5B)|67.0|72.3|44.1|70.6|53.4
>
> ---
>
> >**Q7:**
> Did authors try other well-known code embedding methods, like UniXcoder?
>
> **A7:**
> The training results of Qwen-1.5B using various embedding models are as follows:
> | |Size|CodeNet|F2STrans
> -|-|-|-
> BGE|110M|84.3|68.8
> UniXcoder|123M|85.2|70.1
> Jina-embeddings|161M|85.1|69.8
> CodeXEmbed|400M|85.4|70.5

---

### Official Review · Reviewer_Lczm · 2025-03-26

**Overall Recommendation:** 3

**Summary:**

Previous work in code translation has focused on improving the performance of LLM-based code translation by focusing on multilingual training and various test-time inference strategies. In this paper, the authors hypothesize that optimizing for program correctness and program readability can improve the performance of LLMs on code translation tasks. Specifically, the authors mine codeforces to collect a dataset of cross-lingual code pairs that are functionally consistent and present a contrastive learning framework to optimize for coding style. The authors find that (1) evaluating on their curated benchmark reduces mean average score by about 10% highlighting greater complexity (2) ablating the style-consistent training and the functional consistency checks significantly reduce the overall effectiveness of the model, (3) for many tasks, a 0.5B parameter Quen model outperforms GPT4 model.

**Claims And Evidence:**

Generally yes. I think the authors have presented reasonable evidence that supports their claims. Specifically, I looked at these claims:
 - **Claim 1:** F2STrans is an effective benchmark for code translation.
 - Evidence 1: The mean average success rate of models drops 10% from CodeNet to F2STrans.
Comments: The performance of Python translation drops signficiantly from CodeEval to FS2Trans. The authors mention that this could be because python is an interpreted language but a deeper answer would be useful here. what is the mean average success rate ignoring Python translation tasks?
 - **Claim 2**: Functional consistency training is beneficial to code translation.
 - Evidence 2: An ablation experiment shows that, without functional consistency training, translation success rate drops slightly.
 - **Claim 3:** Style-oriented training is beneficial to code translation.
 - Evidence 3: An ablation showcases that model accuracy suffers without style oriented training.

**Essential References Not Discussed:**

.

**Experimental Designs Or Analyses:**

For Table 2:
> L258R: As shown in Table 2(II), all evaluated models achieve scores that are at least 10 points lower on average in our benchmark compared to CodeNet, highlighting the greater complexity of our proposed benchmark

This seems to be the only quantitative evidence in the main paper that argues for whether FS2Trans is a better benchmark than CodeNet. This is a major claim which deserves a finer-grained analysis than the current analysis presented. Some recommendations:
 - Please present concrete numbers of the mean average success rate delta, ideally by each language category.
 - It seems that Python translation tasks have an outsized influence on the aggregate number presented. If we remove Python, what does the delta difference look like?

> L258R: ... weaker performance on Python...

Its pretty interesting that Python translation success rate decreases so much between the datasets. The reasoning presented in the paper is that this is because of the "interpreted nature [of Python]". I'm not sure this argument holds up very well. If this argument were true, CodeEval results should also have been in the 25%-55% range. Yet, on CodeEval, the average performance is between 70%-90%.  I think a deeper qualitative analysis would be beneficial here, such as a qualitative analysis of the code and a study of what kinds of failures the models fell into on CodeEval and on F2STrans.

__Overall__: For Table 2, I agree that the overall reduction in success rate is interesting. However, there are many other non-trivial insights we can take from this experiment that are not presented. A deeper analysis is required.

Additionally, both CodeEval and FS2Trans source their data from codeforces. I do not think the code-writing style on codeforces is a good indicator of how well code translation would work "in the wild." For instance, we generally modularize and structure code in Java around certain object oriented classes while in C we modularize and structure code around shared procedures. I think evaluating and comparing on a code translation dataset that sources it's data from other sources would be very useful here. (eg: CodeTransOcean but the authors might know other ones as well).



> While the effectiveness of LLMs ... most models face two critical limitations .. (i) Correctness [and] (ii) Readability

Table 2 and related experiments seem to only measure the correctness of the generated code against a dataset. How is code quality improved by the presented approach? How can this be measured quantitatively? Even some qualitative examples of how the code correctness and readability changes before and after training would be beneficial.

**Methods And Evaluation Criteria:**

This paper is generally missing a comparison with the CodeTransOcean benchmark. Evaluating how the trained methods performed on this dataset would be very informative since the CodeEval seems to be sourcing data from the same underlying data distribution as F2STrans (codeforces).

Furthermore, I find it concerning that the data mining algorithm for both the functional consistency and the style-guidance seem to rely heavily on having an LLM as a critic (Quen-7B as the "LLM Judge" and Quen-32B as the "Style Aware LLM") but there was no evaluation done as to how the inherent biases of these models affects the final performance of a model training on the F2STrans benchmark.

**Other Comments Or Suggestions:**

Overall, I'm leaning towards Rejection (Weak Rejection). While F2STrans presents an interesting hypothesis (that style consistency and functional consistency are important for code translation), the supporting analysis for the experimental observations is inadequate to support these claims. I'm willing to raise my score after discussion with the authors.

----
After discussions with the Authors, I'm increasing my score to Weak Accept.

**Other Strengths And Weaknesses:**

- The notation for denoting LLM prompts $\mathcal{P}(\cdot)$ looks very similar to $P(\cdot)$. It might be better to express these as *serializations* of various variables. e.g: instead of using $\mathcal{P}(src),tgt_{<i}$ to denote the "prompt" for translating source code with the text generated so far, we can instead denote it as $\langle \text{src} \rangle;\langle\text{tgt}_{<I}\rangle$ to denote the serialization of the source code concatenated with the target code generated so far.

 - > L63: Relying solely on the inherent capabilities of LLMs to overcome these issues is only a short-term solution.
    - I can agree with this statement but more justification for this statement (in what use cases? Under what constraints?) would be useful.

 - In the related work section, it would be beneficial to comment on how the proposed pipeline relates to each of the sub-fields mentioned.

**Questions For Authors:**

(merged with other sections)

**Relation To Broader Scientific Literature:**

Just in terms of the benchmark, I don't think the results with the style guidance and functional consistency are particularly surprising. I definitely see this benchmark complementing other preexisting benchmarks but I'm not sure what additional non-trivial insight a user would gain by benchmarking their model on CodeEval as compared to F2STrans, other than the fact that their model will be slightly worse on FS2Trans. I believe this can be solved by improving the presentation of the paper to motivate the hypothesis. a bit more. For instance, by showcasing a non-trivial tasks where function consistency and style-oriented training truly brings a large difference in performance.
In terms of the algorithmic insight, the training methodology and loss functions do not seem to be novel contributions.

**Theoretical Claims:**

none.

---

> ### Author Rebuttal · Authors · 2025-04-01
>
> Thank you for recognizing the contributions of our work and providing valuable feedback.
> We respond to each comment as follows and sincerely hope that our rebuttal could properly address your concerns.
> If so, we would deeply appreciate it if you could raise your score (currently Rating: 2: Weak reject).
> If not, please let us know your concerns, and we will continue actively responding to your comments and improving our submission.
>
> Before addressing your specific points, we would like to clarify that the "CodeEval" mentioned in the review likely refers to the "CodeNet" benchmark.
> >**Q1:**
> In addition to the CodeNet and F2STrans benchmarks, the paper should evaluate performance on a broader range of benchmarks, such as CodeTransOcean.
>
> **A1:**
> Besides the CodeNet and F2STrans benchmarks, we also report results on xCodeEval and CodeScope benchmarks in **Table 5 of Appendix D**.
> Here, we show more results on CodeTransOcean and Multiple-E benchmarks.
> For CodeTransOcean, we adopt the CodeBLEU metric, consistent with the standard practice in the CodeTransOcean paper.
>
> ||xCodeEval|CodeScope|Multiple-E|CodeTransOcean
> -|-|-|-|-
> Qwen32B|70.5|58.4|59.9|27.5
> GPT4|77.1|64.6|68.0|29.0
> Ours (Qwen1.5B)|**85.3**|**73.9**|**74.2**|**29.7**
> ---
> >**Q2:**
> The paper lacks evaluation of code quality.
>
> **A2:**
> In **Table 4 of Section 3.5**, we assess the stylistic quality of translations based on the CCSim metric.
> The results are as follows:
>
> ||CSSim
> |-|:-:|
> Qwen32B|76.4
> GPT4|77.9
> Ours (Qwen1.5B)|**82.0**
> ---
> >**Q3:**
> The paper uses Qwen7B and Qwen32B for LLM Judge and Style Aware LLM Translation, respectively. It does not analyze the impact of inherent biases in these LLMs on the model training.
>
> **A3:**
> We build training datasets based on different LLMs (Qwen-7B/32B and DeepSeek-6.7B/33B) and show the training results of Qwen1.5B based on F2STrans benchmark:
>
> LLMs Used by Judge and Style Aware Translation|C|C++|Go|Java|Py|Avg.
> -|:-:|:-:|:-:|:-:|:-:|:-:|
> Qwen-(7B/32B)|81.0|69.2|75.1|83.6|40.0|69.8
> DeepSeek-(6.7B/33B)|80.4|67.1|77.3|82.2|43.4|70.1
> Hybrid Mining|**81.3**|**70.5**|**77.8**|**84.1**|**43.6**|**71.5**
>
> We find that the inherent biases of LLMs cause uneven performance across languages.
> By adopting a **Hybrid Mining** strategy—using Qwen LLMs for C, C++, and Java, and DeepSeek LLMs for Go and Python—we achieved consistent performance improvements.
> This demonstrates that assigning tasks according to each model's strengths can alleviate the impact of LLMs’ inherent biases and improve the quality of training data.
>
> ---
> >**Q4:**
> Some non-trivial insights into the overall performance decline on the F2STrans benchmark are required.
>
> **A4:**
> We have identified two main reasons for the overall performance decline:
>
> - **Number of Test Cases.**
> F2STrans includes 50 human-annotated test cases for each code snippet (see Table 1), whereas CodeNet provides only one.
> The scarcity of test cases in CodeNet may lead to inflated evaluation results, introducing evaluation errors.
> To quantify this impact, we conducted 10 code translation evaluations, each using a random test case from F2STrans samples.
> The results, shown below, reveal that relying on a single test case introduces at least a 2.4% evaluation error.
>
> ||1 Case|50 Cases|Evaluation Error
> |-|:-:|:-:|:-:|
> Qwen32B|56.7|53.9|3.8
> GPT4|66.0|63.6|2.4
> Ours (Qwen3B)|76.1|73.7|2.4
>
> - **Code Length.**
> As shown below, source code in F2STrans usually contains more tokens than in CodeNet, which makes evaluations more challenging.
>
> ||C|C++|Go|Java|Py
> |-|:-:|:-:|:-:|:-:|:-:|
> CodeNet|298.5|264.8|300.6|256.0|91.3
> F2STrans|397.3|368.7|350.6|315.2|216.4
> Delta|98.8|103.9|50.0|59.2|125.1
> ---
> >**Q5:**
> The Python translation performance of various models shows a significant decline in the F2STrans benchmark, yet the paper lacks an analysis.
>
> **A5:**
> We find that Python code in our F2STrans benchmark is more complex, characterized by longer average length and more frequent use of built-in APIs, such as "map" and "lambda".
> As a result, models are more prone to generating code with compilation errors when translating these complex examples.
> For instance, over half of the errors produced by Qwen32B are due to compilation failures, as follows:
>
> ||CodeNet|F2STrans
> -|:-:|:-:|
> #Token|91.3|216.4
> #API|4.3|7.5
> Compilation Error Ratio|21.4|51.8
>
> ---
> >**Q6:**
> Presenting concrete numbers for the mean average success rate delta, both with and without Python, can help verify the benchmark advantages of F2STrans.
>
> **A6:**
> The table below shows the mean average success rate delta for each language between the CodeNet and F2STrans benchmarks.
> Each model's performance drops by at least 11.7 on F2STrans compared to CodeNet, and by at least 6.0 even without Python.
> This confirms that our F2STrans benchmark is highly challenging.
> ||C|C++|Go|Java|Py|Avg.|Avg. w/o Py
> |-|:-:|:-:|:-:|:-:|:-:|:-:|:-:|
> Qwen32B|12.9|26.6|8.7|4.0|42.8|19.0|13.0
> GPT4|5.8|13.6|2.6|1.9|34.4|11.7|6.0
> Ours (Qwen1.5B)|6.4|17.8|5.9|1.7|44.5|15.3|8.0

---

> > ### Comment · Reviewer_Lczm · 2025-04-02
> >
> > Thank you for the additional insights! Apologies about the lack of proofreading on my end. Generally, These additional observations were extremely insightful and I'd recommend adding these to the main paper (if possible). I'm raising my score to Weak Accept.

---

> > > ### Author Response · Authors · 2025-04-03
> > >
> > > Thank you very much for your thoughtful feedback and for raising your score to 3! We deeply appreciate the time and effort you have invested in carefully reviewing our work and providing such constructive suggestions. Your insights have been invaluable in helping us strengthen our paper and address key concerns more comprehensively.
> > >
> > > We will make sure to incorporate the additional results and discussions into the camera-ready version, as you suggested. Your feedback has not only improved the quality of our work but also deepened our understanding of the methods and their implications.
> > >
> > > Once again, thank you for your support and for recognizing our efforts. We hope that our work can contribute meaningfully to the field.

---

### Official Review · Reviewer_p79m · 2025-03-30

**Overall Recommendation:** 3

**Summary:**

Authors present F2STRANS, a novel function-to-style guiding paradigm that enhances the code translation performance of smaller LLMs by addressing functional correctness and stylistic readability.

Authors propose a 2-stage learning approach:
1) The functional learning stage is based on instruction fine tuning(FT) using good quality source-target code pairs collected by authors from online programming platforms (codeforces). This step optimizes for code translation correctness by FT a base LLM. The quality of code pairs is maintained by (a) Qwen-7B as LLM judge (b) testing.
2) The style learning stage improves code readability by training model fine-tuned in stage 1 using constrastive learning (positive and negative style examples). The positive example is generated by Qwen-32B while negative style translations are generated from model fine-tuned in stage 1.

Authors present extensive evaluations which shows that F2STRANS significantly improves code translation performance across benchmarks for 4 different prompting approaches based on FT smaller Qwen and starcoder models and comparing these against larger Qwen-32B and GPT-4.

Authors also introduces a new larger code translation benchmark F2STRANS.

**Claims And Evidence:**

Key technical Innovations claimed include:
1. Relevance-driven code pair selection:
(a) Use of Jina code embedding model & LLM judge to assess solution consistency
(b) Differential testing ensures code pairs exhibit identical input-output behavior
2. Style consensus selection:
a) Leverages CSSim metric to ensure stylistic similarity among candidates translations.
3. New code translation benchmark dataset with 5 programming languages.

Main results claimed include:
1. F2STRANS improves translation performance on both the new (fs2trans) and existing benchmarks (codenet). F2STRANS enables smaller Qwen and starcoder models to outperform Qwen32B and GPT-4 on several language pairs.
2. Ablation studies demonstrate that functional and style learning stages contribute substantially to performance gains. F2STRANS shows improved effectiveness in correcting various types of errors in base LLM translations compared to function guidance alone.
3. Evaluation using CCSim metric shows F2STRANS enhances stylistic quality of translated code.
4. Experiments with multilingual training strategies indicate that cross-lingual training enhances the performance of F2STRANS.

**Essential References Not Discussed:**

NIL

**Experimental Designs Or Analyses:**

NIL

**Methods And Evaluation Criteria:**

Yes . Authors use (i) a new larger benchmark (ii) CCSim to evaluate the stylistic similarity (iii) Comparison with stronger LLMs QWEN-32B and GPT-4. (iv) Ablation Studies to attribute the contribution of different components of FS2TRANS. (v) Analysis of types of errors corrected by F2STRANS.

**Other Comments Or Suggestions:**

None

**Other Strengths And Weaknesses:**

- Weakness: Heavy dependence on the availability of larger LLMs to judge translations and generate correctly styled translations.

**Questions For Authors:**

- Could you please explain how the negative-style translations are generated using Mfun in stage 2 along with prompting techniques used.
How did you validate that there is sufficient variability/diversity in the generated negative style samples and that they are different from one another.

**Relation To Broader Scientific Literature:**

Broadly, this work makes a good contribution in the code translation space, by building upon ideas from literature and contributing a new benchmark.

**Theoretical Claims:**

None

---

> ### Author Rebuttal · Authors · 2025-04-01
>
> We thank the reviewer for the insightful and valuable comments.
> We respond to each comment as follows and sincerely hope that our rebuttal could properly address your concerns.
> If our response meets your expectations, we would greatly appreciate it if you could consider raising your score (currently Rating: 3: Weak Accept).
> If further concerns remain, please let us know, and we are committed to addressing them and refining our submission accordingly.
>
>
> >**Q1:**
> Heavy dependence on the availability of larger LLMs to judge translations and generate correctly styled translations.
>
>
> **A1:**
> Using larger LLMs to build training data can indeed enhance model performance.
> However, we would like to clarify two key points:
>
> (1) **While it is important to acknowledge the limitations of this approach, we should not overlook its significant benefits.**
>
> For instance, our training framework enables Qwen-1.5B to outperform GPT-4 across a wide range of code translation benchmarks.
> The specific results are as follows:
>
> ||CodeNet|F2STrans|xCodeEval|CodeScope|
> |-|:-:|:-:|:-:|:-:|
> |Qwen-32B| 72.9 | 53.9 |  70.5 | 58.4 |
> |GPT-4| 75.3 | 63.6 | 77.1 | 64.6 |
> |Ours (Qwen-1.5B)| **85.1** |  **69.8** |  **85.3** | **73.9** |
>
>
> (2) **Even when we use LLMs of the same scale as the trained model to construct training data, our two-stage training method remains effective.**
>
>
> Here, we trained Qwen-3B exclusively on data constructed entirely by itself.
> The results are presented in the table below.
> We find that, without relying on larger LLMs, our framework still enables Qwen-3B to surpass GPT-4 across these code translation benchmarks.
>
>
> ||CodeNet|F2STrans|xCodeEval|CodeScope|
> |-|:-:|:-:|:-:|:-:|
> |Qwen-32B| 72.9 | 53.9 |  70.5 | 58.4 |
> |GPT-4| 75.3 | 63.6 | 77.1 | 64.6 |
> |Ours (Judge-**Qwen-3B**, Positive Translation Generation-**Qwen-3B**, Model Training-**Qwen-3B**)| **85.5** | **70.3** | **85.7** | **75.1** |
>
> ---
>
> >**Q2:**
> Could you please explain how the negative-style translations are generated using $M_{fun}$ in stage 2 along with prompting techniques used.
> How did you validate that there is sufficient variability/diversity in the generated negative style samples and that they are different from one another.
>
>
> **A2:**
> We set $M_{fun}$'s temperature to 1.5 and perform multiple sampling generations with the prompt described in line 825 of our paper.
> A translation is retained only if its CCsim score with any prior translation is **below 0.9**; otherwise, it is discarded.
> CCSim, as detailed in lines 606 to 633 of the paper, measures stylistic similarity between code snippets, and a threshold of 0.9 ensures that the retained samples are sufficiently distinct from one another.
> If fewer than the desired number of translations are retained after 20 sampling attempts (a rare scenario), we incrementally increase the temperature to encourage greater diversity in the $M_{fun}$'s outputs.
> The table below illustrates the average similarity between our constructed negative-style translations, measured using two metrics: CCSim and CodeBLEU.
>
> |         | C    | C++  | Go   | Java | Python | Average |
> |---------|:------:|:------:|:------:|:------:|:--------:|:---------:|
> | CCSim   | 82.4 | 84.3 | 77.5 | 81.7 | 73.6   | 79.9    |
> | CodeBLUE| 78.1 | 79.7 | 72.1 | 76.4 | 68.7   | 75.0    |
>
>
>
>
> We further evaluated the performance of the trained Qwen-1.5B model on various benchmarks using different CCsim filtering thresholds.
> The results shown in the following table indicate that omitting negative translation filtering—disregarding the diversity of negative translations—led to a noticeable decline in performance.
> Moreover, setting the threshold too low (e.g., 0.8) also impaired performance.
> This may be because an excessively low threshold introduces some highly diverse but low-quality negative translations.
>
>
> | Benchmark       | 1.0 (w/o Negative Translation Filtering) | 0.95   | 0.9    | 0.85   | 0.8    |
> |-----------------|:------------------------------------:|:------:|:------:|:------:|:------:|
> | CodeNet     |                84.0                   |  84.5  | **85.1** | **85.1** |  84.9  |
> | F2STrans    |               68.9                  |  69.3  | **69.8** |  69.7  |  69.5  |
> | xCodeEval   |               84.6                  |  84.8  | 85.3 | **85.4**  |  85.1  |
> | CodeScope   |               73.1                  |  73.5  | **73.9** |  73.7  |  73.4  |

---

### Decision · Program_Chairs · 2025-05-01

**Decision:**

Accept (poster)

**Comment:**

The paper gives a new approach to LLM-based code translation. The method operates in two stages. First, instruction finetuning is used to translate code into a functionally correct form in the target language. Second, a contrastive learning method converts this functionally correct translated code into stylistically idiomatic code. The paper evaluates the method rigorously and also introduces a new benchmark.

The reviewers were generally in favor of accepting the paper. There were a few questions (most notably about alternative baselines and analysis), but the reviewers have answered the questions diligently and provided additional experimental evidence in their rebuttal. Given this, I am recommending acceptance. Please incorporate the reviewers' feedback and the new results carefully in the final version.